# A doubly stochastic renewal framework for partitioning spiking variability

Cina Aghamohammadi [1,2], Chandramouli Chandrasekaran [3,4,5,6] &
Tatiana A. Engel [1,2] ✉

The firing rate is a prevalent concept used to describe neural computations, but estimating dynamically changing firing rates from irregular spikes is challenging. An inhomogeneous Poisson process, the standard model for partitioning firing rate and spiking irregularity, cannot account for diverse spike statistics observed across neurons. We introduce a doubly stochastic renewal point process, a flexible mathematical framework for partitioning spiking variability, which captures the broad spectrum of spiking irregularity from periodic to super-Poisson. We validate our partitioning framework using intracellular voltage recordings and develop a method for estimating spiking irregularity from data. We find that the spiking irregularity of cortical neurons decreases from sensory to association areas and is nearly constant for each neuron under many conditions but can also change across task epochs. Spiking network models show that spiking irregularity depends on connectivity and can change with external input. These results help improve the precision of estimating firing rates on single trials and constrain mechanistic models of neural circuits.

The vast array of diverse brain functions arises from irregular spiking across neural populations, observable on recording electrodes in experiments. Yet, most current hypotheses about neural computation employ the firing rate, an abstract mathematical quantity describing the propensity of a neuron to spike. For example, prevalent theories suggest that neurons change their firing rate to signal features of sensory stimuli[1,2] or parameters of body movements[3]. Moreover, complex cognitive computations emerge from the dynamics of latent variables tracing trajectories through a state space, where each axis corresponds to the firing rate of one neuron in the population[4–9]. Since the firing rate is a latent quantity not directly observable in experiments, testing any of these theories requires relating the firing rate to experimentally measured spikes. Specifying this relationship effectively partitions the total variability in the spiking output of a neuron into two components: changes in the firing rate and irregularity of the process generating spikes from this firing rate[10,11].

The traditional partitioning method defines the firing rate as the average of spikes over repeated trials under the same experimental conditions, thus attributing any trial-to-trial variability entirely to the irregular spike generation in single neurons[1,2,4]. This definition implies that the firing rate is the same on each trial and deterministically locked to experimentally controlled events, e.g., stimulus or movement onset. However, neural responses fluctuate significantly across trials due to many factors beyond experimental control, including changes in behavioral state[12–14] and endogenous network dynamics[10,15,16]. Capturing the dynamics of these latent factors requires a partitioning framework that accounts for the firing rate fluctuations on single trials.

The most common approach to account for the firing rate fluctuations is to assume an inhomogeneous Poisson process as a model for spike generation on single trials[11]. In this model, spikes occur independently of each other, with the probability of spiking at each

[1]Princeton Neuroscience Institute, Princeton University, Princeton, NJ, USA. [2]Cold Spring Harbor Laboratory, Cold Spring Harbor, NY, USA. [3]Department of Anatomy & Neurobiology, Boston University, Boston, MA, USA. [4]Department of Psychological and Brain Sciences, Boston University, Boston, MA, USA. [5]Department of Biomedical Engineering, Boston University, Boston, MA, USA. [6]Center for Systems Neuroscience, Boston University, Boston, MA, USA. ✉e-mail: tatiana.engel@princeton.edu

moment controlled by the instantaneous firing rate[17]. The mathematical convenience of the inhomogeneous Poisson process led to its widespread application in existing methods for inferring the dynamics of firing rates and associated latent variables on single trials[18–25]. However, the spiking of neurons across many brain areas deviates significantly from the inhomogeneous Poisson assumption. For example, many neurons have a Fano factor (FF, the variance-to-mean ratio of spike counts) less than one[10,26–28], which is the minimal FF value theoretically possible for an inhomogeneous Poisson process. Unfortunately, incorrect assumptions about the irregularity of the spiking process can lead to errors in estimating the link between changes in firing rates and behavior and to misleading conclusions when arbitrating between alternative hypotheses about single-trial dynamics of latent variables[29,30]. Despite these limitations, there is no alternative model for accurately partitioning spiking variability in experimental data. While few previous methods attempted to estimate spiking irregularity moving beyond the Poisson assumption[10,28,31], these methods relied on heuristic assumptions that are not always true and therefore, as we show, are sensitive to nuisance parameters, e.g., firing rate and bin size.

We introduce a doubly stochastic renewal (DSR) point process, a mathematical framework for partitioning spiking variability, which accounts for firing rate fluctuations and provides a flexible model to capture the broad spectrum of spiking irregularity from periodic to super-Poisson. Using our framework, we devise a method for estimating spiking irregularity and show that it accurately recovers ground truth on simulated spike trains with a wide range of firing rate dynamics and spiking irregularity. We further validate the theoretical assumptions in our framework using intracellular recordings of membrane potential from neurons in the barrel cortex of awake mice[32]. We apply our approach to quantify the spiking irregularity of cortical neurons and find that spiking irregularity decreases from visual to higher sensory-motor areas, mirroring the gradient of unpartitioned variability[26,27,33,34]. Moreover, the spiking irregularity is nearly constant for each neuron under many conditions but can also change across task epochs. Using spiking network models[35,36], we show that spiking irregularity depends on connectivity and biophysical properties of single neurons and can change with external input. Our work establishes that a DSR point process is a flexible model to capture the broad spectrum of spiking irregularity of cortical neurons and improve the precision of methods for estimating latent dynamics on single trials.

## Results

We develop the DSR point process model as a flexible alternative to the standard inhomogeneous Poisson process, capable of accounting for the diverse spiking irregularity of cortical neurons. We first introduce our mathematical framework and precisely define the spiking irregularity as a parameter $\phi$ within the DSR model. We then present a method for estimating $\phi$ from data and validate its accuracy using synthetic spike trains and intracellular voltage recordings. Finally, we describe our observations of spiking irregularity in several cortical areas, which reveal systematic differences from sensory to motor areas and indicate that Poisson irregularity is a rare exception, not a rule.

### Doubly stochastic renewal framework

Mathematically, a spike train is a point process, that is, a sequence of discrete events occurring randomly in time. A doubly stochastic point process generates spikes stochastically from the firing rate that also fluctuates over time and from trial to trial. To define a doubly stochastic point process, we need to specify two components: a non-negative real-valued stochastic process $\lambda(t)$ for the instantaneous firing rate and a point process generating spikes from a realization of the firing rate $\lambda(t)$.

The simplest point process model is the Poisson process[17], in which spikes occur independently of each other with the probability $\lambda(t)dt$ in

any infinitesimal time interval $[t, t + dt]$. The Poisson point process generates spikes with fixed irregularity: for a constant firing rate, the FF equals one for a time bin of any size. Since firing rate fluctuations only increase variability[10], FF is always greater than one for Poisson processes with fluctuating firing rate[11]. Therefore, the inhomogeneous Poisson process cannot account for diverse spiking statistics across neurons, in particular, neurons with FF smaller than one[10,26–28].

A more flexible model is a renewal point process, in which the probability of generating a spike depends on the time elapsed since the last spike[17,37]. Mathematically, the dependence between consecutive spike times can be described by the probability density $g(\cdot)$ of interspike intervals (ISIs). For the constant firing rate, the probability of the next spike occurring in the interval $[t, t + dt]$ is proportional to $g(t - t_1)dt$, where $t_1$ is the time of the last spike. The shape of the ISI distribution $g(\cdot)$ controls the irregularity of the renewal point process. A narrow distribution peaked around a set ISI value will produce a nearly periodic spike train, whereas an exponential distribution results in an irregular spike train equivalent to the Poisson process.

Here, we introduce a DSR point process, which we define by a pair $\{g(\cdot), \lambda(t)\}$ via a three-step algorithm for generating spike trains (Fig. 1a). First, we sample a realization of the firing rate $\lambda(t)$ for a specific trial. Next, we sample ISIs from the probability density $g(\cdot)$ to generate spikes in operational time $t'$. We set the mean of the ISI probability density function to one: $\mu_g = \int_0^\infty \theta g(\theta)d\theta = 1$ s, which implies the firing rate of the point process in operational time is $\mu_g^{-1} = 1$ Hz. Finally, we map the spike times from the operational time $t'$ to the real time $t$ by locally squeezing and stretching time in proportion to the inverse cumulative firing rate $t = \Lambda^{-1}(t')$, where

$$t' = \Lambda(t) = \int_0^t \lambda(s)ds. \tag{1}$$

This transformation ensures that the spike density in real time follows the instantaneous firing rate $\lambda(t)$[38,39].

In our framework, the spiking irregularity is defined by the ISI distribution $g(\cdot)$ in the operational time, which controls spiking irregularity independently of the firing rate. In particular, using the same distribution $g(\cdot)$ (the same spiking irregularity), our DSR model can generate spike trains with high or low firing rate using different $\lambda(t)$. Conversely, for the same firing rate, our DSR model can generate spike trains with high or low spiking irregularity using different distributions $g(\cdot)$. A special case is when $g(\cdot)$ belongs to a two-parameter family of continuous probability distributions uniquely determined by its mean $\mu_g$ and standard deviation $\sigma_g$. In this case, we denote the squared coefficient of variation[40] (CV²) of the distribution $g(\cdot)$ by

$$\phi \equiv \frac{\sigma_g^2}{\mu_g^2}. \tag{2}$$

With these assumptions, $\phi$ uniquely determines the distribution $g(\cdot)$, since $\mu_g = 1$ s by our definition of the DSR process. Therefore, a single parameter $\phi$ fully controls the spiking irregularity. For different values of $\phi$ and firing-rate fluctuations, our DSR model can generate a broad spectrum of spiking activity, ranging from nearly periodic to highly irregular both within and across trials (Fig. 1b).

### Partitioning variability in data

In experiments, we only have access to the total spiking variability that includes contributions from both firing rate fluctuations and spiking irregularity. A common metric of the total spiking variability is the variance Var($N_T$) of spike count $N_T$ measured in time bins of size $T$. For doubly stochastic processes, the total variance Var($N_T$) arises from the firing rate and point process components. We assume that the instantaneous firing rate changes on a timescale $\tau$ longer than the bin size $\tau > T$, which implies $\lambda(t)$ is approximately constant $\lambda$ within a bin.

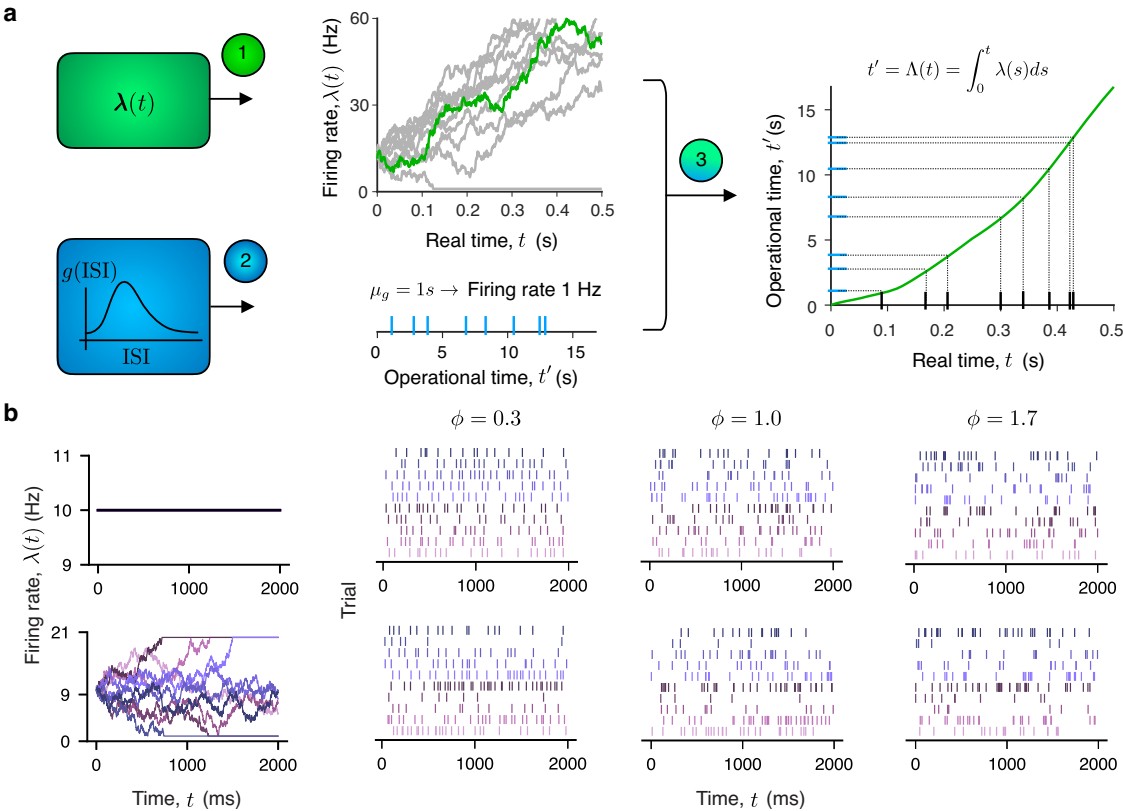

**Fig. 1 | Doubly stochastic renewal point process model. a** We define a doubly stochastic renewal point process by a pair $\{g(\cdot), \lambda(t)\}$. A neuron-specific function $g(\cdot)$ is the ISI probability density in the operational time $t'$ which controls the irregularity of the renewal point process, that is, the variability of spike generation (lower left, blue). A stochastic process $\lambda(t)$ defines the dynamics of firing rate on single trials and controls the trial-to-trial firing rate fluctuations (upper left, green). The pair $\{g(\cdot), \lambda(t)\}$ defines the spike generation process via a three-step algorithm. First, a realization of the firing rate $\lambda(t)$ for a specific trial is sampled from the process $\lambda(t)$ (upper center, green$-\lambda(t)$ for a specific trial, gray$-\lambda(t)$ for multiple trials). Second, spike times are sampled in the operational time from the ISI probability density function $g(\cdot)$ (lower center, blue ticks). Since the mean of the ISI distribution $g(\cdot)$ is set to $\mu_g = 1\,$s, the mean firing rate of spikes in the operational time is 1 Hz. Third, the spikes are mapped from the operational time $t'$ to the real time $t$ via $t = \Lambda^{-1}(t')$, where the map is defined by the cumulative firing rate function $t' = \Lambda(t) = \int_0^t \lambda(s)ds$ (right, green line). **b** Examples of diverse spiking activity generated from a doubly stochastic renewal model. We consider firing rate $\lambda(t)$ that is constant within and across trials (upper row) or follows a drift-diffusion process with sticky boundaries on single trials (lower row). From each realization of the firing rate $\lambda(t)$ (color gradient, first column), we generate spikes using the doubly stochastic renewal model in which $g(\cdot)$ is a gamma distribution with $\phi = 0.3$ (sub-Poisson, second column), $\phi = 1$ (Poisson, third column), and $\phi = 1.7$ (super-Poisson, fourth column). Differences in firing rate variability and spiking irregularity $\phi$ are difficult to discern from the resulting diverse patterns of spiking activity.

Then, we can use the law of total variance[41] to decompose the total spike-count variance into the firing rate and point process components[10]:

$$\text{Var}(\boldsymbol{N}_T) = \underbrace{\text{Var}(\text{E}[\boldsymbol{N}_T|\boldsymbol{\lambda}])}_{\text{firing rate variance}} + \underbrace{\text{E}[\text{Var}(\boldsymbol{N}_T|\boldsymbol{\lambda})]}_{\text{point process variance}}. \quad (3)$$

Within our DSR framework, we can express the two terms in this decomposition via $\{g(\cdot), \boldsymbol{\lambda}\}$. The first term is the firing rate variance and equals $\text{Var}(\text{E}[\boldsymbol{N}_T|\boldsymbol{\lambda}]) = \text{Var}(\lambda T)$ (Methods). The second term is the point process variance and, for moderately large bin size $T > 1/\text{E}[\boldsymbol{\lambda}]$, we can approximate it as (Supplementary Note 1.1):

$$\text{E}[\text{Var}(\boldsymbol{N}_T|\boldsymbol{\lambda})] = \left(\frac{\sigma_g}{\mu_g}\right)^2 \text{E}[\boldsymbol{N}_T] + \frac{1}{6} + \frac{1}{2} \cdot \left(\frac{\sigma_g}{\mu_g}\right)^4 - \frac{1}{3} \cdot \frac{\mu_{3g}}{\mu_g^3} + \mathcal{O}(T^{-1}). \quad (4)$$

Here $\mu_g$, $\sigma_g$, and $\mu_{3g}$ are the mean, standard deviation, and third central moment of the distribution $g(\cdot)$, and $\mathcal{O}(T^{-1})$ indicates the approximation error scaling as $T^{-1}$. Since $\mu_g = 1\,$s by our definition of the DSR process, two parameters $\sigma_g$ and $\mu_{3g}$ control the point process variance.

Next, we introduce simplifying assumptions to develop this general theoretical result into a practical data analysis method. We

consider $g(\cdot)$ to be the gamma distribution, a particular case of the two-parameter distribution family that has proven useful for modeling ISI data[38,42,43]. For the gamma distribution, the third central moment is given as $\mu_{3g} = 2\phi^2$ (Methods), and we can simplify the partitioning equation Eq. (4) to be

$$\text{Var}(\boldsymbol{N}_T) = \text{Var}(\boldsymbol{\lambda}T) + \frac{1}{6}(1 - \phi^2) + \phi\,\text{E}[\boldsymbol{N}_T] + \mathcal{O}(T^{-1}). \quad (5)$$

The second and third terms in this equation express the point process variance via a single parameter $\phi$. For $\phi = 1$, the gamma distribution reduces to the exponential distribution, and the renewal point process is a Poisson process with variance equal to the mean spike count in Eq. (5).

In summary, we partitioned the total spike-count variance $\text{Var}(\boldsymbol{N}_T)$ into the firing rate and point process components, with spiking irregularity controlled by a single parameter $\phi$. To make this partition unambiguous[29], we constrained the spike generation to be a renewal point process and enforced the smoothness of the firing rate.

### Estimation from data
In partitioning equation Eq. (5), the spike-count mean $\text{E}[\boldsymbol{N}_T]$ and variance $\text{Var}(\boldsymbol{N}_T)$ can be measured directly from data, whereas $\phi$ and the

firing rate variance Var($\lambda T$) are unknown. Thus, to partition variability in spike data, we first need to estimate $\phi$. We devise an estimation method for $\phi$, which we call the DSR method, based on our assumption that the firing rate changes smoothly and is approximately constant within a bin. We apply Eq. (5) to spike counts measured in two bin sizes, $T$ and $2T$, to yield two equations, which we solve to obtain a quadratic equation for $\phi$ (Methods):

$$\frac{1}{2}\left(\phi_{\text{DSR}}^2 - 1\right) - (4\,\text{E}[\boldsymbol{N}_T] - \text{E}[\boldsymbol{N}_{2T}])\phi_{\text{DSR}} + 4\,\text{Var}(\boldsymbol{N}_T) - \text{Var}(\boldsymbol{N}_{2T}) = 0\,. \tag{6}$$

Here, the spike-count mean and variance for each bin size $\text{E}[\boldsymbol{N}_T]$, $\text{E}[\boldsymbol{N}_{2T}]$, $\text{Var}(\boldsymbol{N}_T)$, $\text{Var}(\boldsymbol{N}_{2T})$ are measured directly from the spike data, and $\phi_{\text{DSR}}$ is the only unknown variable. Thus, we can solve Eq. (6) to estimate $\phi_{\text{DSR}}$ from data. Then, we can separate the firing rate and point process variance using the estimated $\phi_{\text{DSR}}$ in Eq. (5).

We consider two criteria for selecting the bin size $T$ used to estimate spiking irregularity. On one hand, we require $T > 1/\text{E}[\lambda]$ to ensure that the partitioning Eq. (5) holds. On the other hand, the bin size should be as small as possible, given the assumption that the firing rate remains constant within each bin. To satisfy both conditions, we set $T = 2/\text{E}[\lambda]$ for each neuron in all analyses (Methods). We confirmed that our estimation method remains robust across a broad range of bin sizes (Supplementary Note 1.2, Supplementary Figs. 1–3).

## Comparison with previous estimation methods
We derive our partition equation Eq. (4) and estimation method Eq. (6) rigorously using the theory of renewal point processes[17,37]. Two other methods have been broadly used for estimating point process variance[10,27,28,39,44,45], but these previous methods relied on several assumptions and heuristics that are not always applicable. We first theoretically analyze how these assumptions impact the estimation accuracy of previous methods. We then evaluate these previous approaches and our method on synthetic data with known ground truth.

The first method, which we refer to as the deterministic time rescaling (DTR) method[39], assumes that the time-dependent firing rate $\lambda(t)$ is deterministic, i.e., does not fluctuate from trial to trial. Accordingly, the firing rate is the same on each trial and can be estimated by averaging spike counts in a bin across trials $\hat{\lambda}(t_i)$, which is called a peristimulus time histogram[46]. One can then substitute the estimated firing rate $\hat{\lambda}(t_i)$ in Eq. (1) to map the spike times into the operational time $t'$. This mapping removes the effect of the time-dependent changes in firing rate locked to experimentally controlled events, assuming that the ISI distribution in the operational time reflects only the point process variability, which corresponds to $g(\cdot)$ in our theory. Accordingly, $\text{CV}^2$ of the rescaled ISIs provides an estimate of the spiking irregularity parameter $\phi$, which we denote by $\phi_{\text{DTR}}$. If the ground-truth firing rate is indeed the same on each trial, $\phi_{\text{DTR}}$ converges to the ground-truth $\phi$ for a large trial number (Methods, Supplementary Note 1.3). However, in the presence of the trial-to-trial firing rate fluctuations, this method always overestimates the point process variability, with the error increasing for larger trial-to-trial variability in the firing rate (Methods, Supplementary Note 1.3, Supplementary Fig. 4).

The second method, to which we refer as the minimum ratio (MR) method[10], allows for firing-rate fluctuations but assumes that the point process variance in Eq. (3) is proportional to the mean spike count:

$$\text{E}[\text{Var}(\boldsymbol{N}_T|\boldsymbol{\lambda})] = \phi\,\text{E}[\boldsymbol{N}_T], \tag{7}$$

where the coefficient $\phi$ is a neuron-specific constant. For renewal processes, this assumption Eq. (7), is only valid in the limit $T \to \infty$ or in the special case when $\mu_{3g} = \frac{3}{2}\phi^2 + \frac{1}{2}$ (Methods, Supplementary

Note 1.4). The latter condition does not hold in general; for example, when $g(\cdot)$ is a gamma distribution, it holds only for $\phi = 1$, i.e., for the Poisson spike generation process (Eq. (5)). Using the ansatz Eq. (7) and the constraint that spike-count variance in Eq. (3) must be positive, the MR method then estimates $\phi$ as the minimum FF across all time bins:

$$\phi_{\text{MR}} = \min_t\left\{\frac{\text{Var}(\boldsymbol{N}_T(t))}{\text{E}[\boldsymbol{N}_T(t)]}\right\}. \tag{8}$$

We show that for DSR processes, the estimation error of this method depends on the bin size $T$, the mean and variance of the firing rate $\lambda(t)$, and on the ground truth $\phi$ itself (Methods, Supplementary Note 1.4). The dependence of $\phi_{\text{MR}}$ on all these nuisance parameters is inconsistent with the assumption that $\phi$ is a constant characterizing the renewal process that controls spiking irregularity. Moreover, it shows that $\phi_{\text{MR}}$ is affected by several sources of bias, leading to unpredictable estimation errors. Other methods for estimating $\phi$ in Eq. (7) with a finite bin size have similar limitations[31,47] (Methods), although accurate estimation is possible using large bin sizes[48].

We compared the performance of our DSR method with these previous methods on synthetic data generated from DSR point processes with known ground truth $\phi$ (Fig. 2). Specifically, we chose $g(\cdot)$ to be a gamma distribution, and the firing rate $\lambda(t)$ either to be constant on each trial sampled from a uniform distribution $[\mu - \frac{w}{2}, \mu + \frac{w}{2}]$ across trials or to follow a drift-diffusion process on single trials as in prominent decision-making models[49]. Since the DTR assumes an inhomogeneous renewal process as a generative model—a special case of our DSR model with zero trial-to-trial firing rate variability—it performs well when this variability is low, in a regime consistent with its assumptions. When trial-to-trial firing rate variability is nonzero, DTR always overestimates $\phi$, with error increasing as the firing rate variability grows. Since the MR method is based on heuristics lacking a generative model, it is not possible to evaluate its accuracy in a setting consistent with its assumptions. Accordingly, the MR method can overestimate or underestimate $\phi$, and the degree of bias depends on the firing-rate variability and the ground-truth $\phi$. In contrast, our DSR method accurately estimates $\phi$, and its accuracy is independent of the firing-rate variability and the ground-truth $\phi$ (Fig. 2c). Thus, the DSR method can reliably estimate $\phi$ for point processes across a wide range of spiking irregularity and firing rate variability.

## Validation with intracellular voltage recordings
After confirming the accuracy of our partitioning method on synthetic data, we sought to validate our theoretical framework on neural recording data. For such validation, extracellular spike recordings are unsuitable because they do not provide an objective, independent measure of instantaneous firing rate on single trials. Instead, we use whole-cell recordings of intracellular membrane potential to estimate the instantaneous firing rate from the subthreshold voltage traces. We base our analysis on theoretical studies showing that, for a variety of leaky integrate-and-fire neuron models, the average firing rate is a power law function of the membrane potential[50,51], which was confirmed experimentally in many cases[52,53]. Accordingly, we model the firing rate of a neuron as a deterministic function of the average subthreshold membrane potential. Using this function, we can obtain the instantaneous firing rate from the subthreshold voltage. With this independently estimated instantaneous firing rate, we can map spikes from real to operational time via time rescaling (Fig. 1a). Since $\phi$ in our framework is defined as $\text{CV}^2 = \sigma_g^2/\mu_g^2$ of the ISI distribution $g(\cdot)$ in operational time, $\text{CV}^2$ of the rescaled ISIs provides an independent estimate of $\phi$ which we can compare to $\phi$ estimated with the DSR method from spike times alone. A good agreement between these different estimates of $\phi$ would indicate that our DSR framework faithfully captures the statistics of biophysical processes relating subthreshold voltage dynamics to spikes.

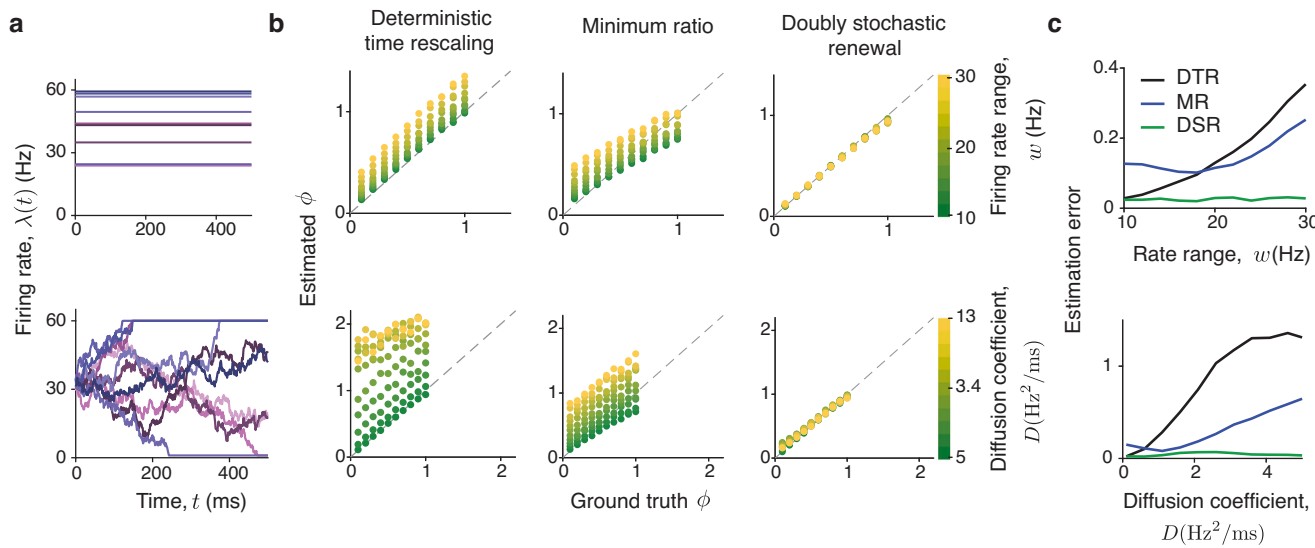

**Fig. 2 | Estimation of spiking irregularity on synthetic data with known ground truth ϕ. a** We generated synthetic data from an ensemble of doubly stochastic renewal point processes $\{g(\cdot), \lambda(t)\}$, where $g(\cdot)$ is the gamma distribution and the ground-truth value of $\phi$ ranges from 0.1 to 1. The firing rate $\lambda(t)$ is either constant within a trial, sampled across trials from a uniform distribution with the width $w$ (upper row), or a drift-diffusion process with sticky boundaries and the diffusion coefficient $D$ (lower row). The parameters $w$ and $D$ control the trial-to-trial variability of the firing rate. We varied $w$ from 10 to 30 Hz, and $D$ from 5 to 13 Hz²/ms. **b** The ground-truth $\phi$ (x-axis) versus estimated $\phi$ (y-axis) for the ensemble of doubly stochastic renewal point processes with uniform (upper row) and drift-diffusion (lower row) firing rate fluctuations. The deterministic time rescaling (DTR) method always overestimates $\phi$ with error increasing for larger trial-to-trial

variability of the firing rate (left). The minimum ratio (MR) method can underestimate or overestimate $\phi$ depending on the mean and variance of the firing rate, bin size, and the ground truth $\phi$ itself, producing unpredictable estimation errors (center). The doubly stochastic renewal (DSR) method accurately estimates $\phi$ in all cases (right). Every point in the scatter plots is the average over 20 simulations for a fixed $\phi$ and fixed $D$ or $w$. Each simulation had 100 trials. **c** Estimation error (root-mean-square error, RMSE, across 20 simulations) for $\phi$ estimated by the three methods for different values of $w$ (upper row) and $D$ (lower row), which control the trial-to-trial variability of the firing rate. The RMSE increases with the firing rate variability for both DTR and MR methods, whereas the RMSE is consistently low and independent of the firing rate variability for our DSR method. Source data are provided as a Source data file.

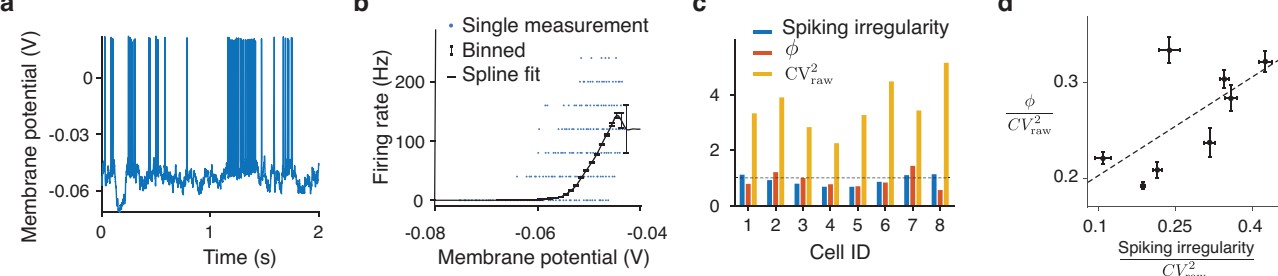

**Fig. 3 | Validation of the DSR framework using intracellular voltage recordings.** **a** Example recording of the membrane potential in a PV neuron on a single trial[32]. The trace contains spikes and subthreshold voltage fluctuations. **b** A relationship between the membrane potential and instantaneous firing rate for the example neuron in (**a**). For each of the 25 ms bins in the recording, we estimate the instantaneous firing rate by the ratio of the spike count to the bin size and plot it against the subthreshold membrane potential averaged over the bin after removing spikes (blue points). Averaging the data points within 1 mV voltage bins reveals a lawful relationship between the subthreshold membrane potential and the firing rate (black points, error bars are s.e.m. over the data points in each voltage bin), which can be approximated with a spline fit (black line). **c** To independently estimate the spiking irregularity of a neuron, we map its spikes to the operational time using the instantaneous firing rate computed from the subthreshold membrane potential at each time using the fitted voltage-to-firing-rate

relationship in (**b**). The spiking irregularity is CV² of ISIs in the operational time (blue) and accounts for a small fraction of the total variability $CV^2_{raw}$ of the ISIs in the real time (yellow), which also contains the firing rate variability. For all 8 neurons in the dataset, $\phi$ estimated with the DSR method from spike times alone (orange) closely corresponds with the spiking irregularity estimated independently from the subthreshold voltage, validating the assumptions of the DSR framework. The dashed line indicates the Poisson irregularity value of 1. **d** The fraction of the total spiking variability $CV^2_{raw}$ attributed to the spiking irregularity was similar between the DSR method (y-axis) and the partitioning based on the subthreshold voltage (x-axis). Each data point represents one neuron. For each neuron, the recording was divided into 20 segments of 10 s duration each. Dots indicate the mean of the estimates across these segments, and error bars show the standard error of the mean (SEM). Source data are provided as a Source data file.

We analyzed a dataset of whole-cell intracellular recordings of the membrane potential in parvalbumin-positive (PV) inhibitory neurons from layer 2/3 of the barrel cortex in awake head-fixed mice[32] (Methods). We analyzed eight neurons from five mice (Fig. 3a). For each neuron, we first computed the empirical relationship between the

average subthreshold voltage and firing rate estimated from spike counts in 50 ms time bins (Fig. 3b, Methods). Consistent with previous studies[52,53], this relationship showed a lawful monotonically increasing trend on average, which we approximated with a smooth deterministic function $f(v)$ by fitting a spline to the data points (Fig. 3b). We assumed

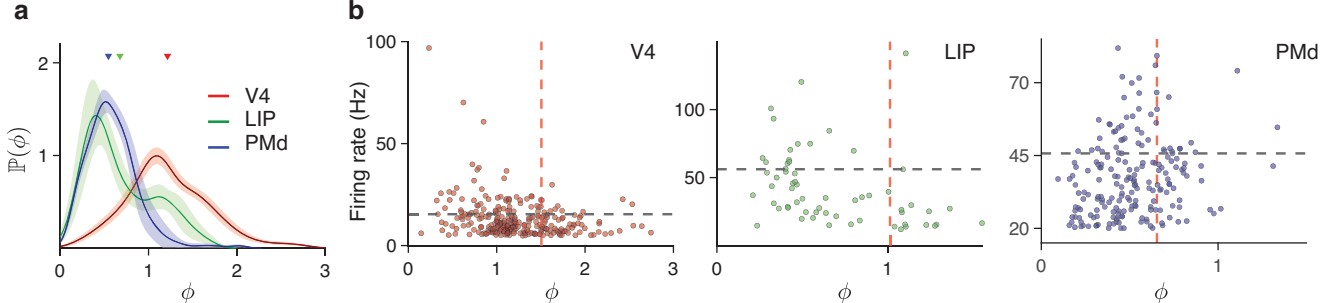

**Fig. 4 | The diversity of spiking irregularity across neurons and cortical areas.** **a** Distributions of spiking irregularity $\phi$ across neurons in cortical areas V4 (red), LIP (green), and PMd (blue). Triangles mark the mean $\phi$ across neurons in each area. The probability densities are computed using a Gaussian kernel density estimator (with kernel widths of 0.15, 0.15, and 0.08 for V4, LIP, and PMd, respectively). Shading indicates the standard deviation across 100 bootstrap samples obtained by resampling neurons. **b** Spiking irregularity $\phi$ (x-axis) versus mean firing rate (y-axis) for each neuron (dots) in V4 (left), LIP (center), and PMd (right). The lines indicate 0.75 quantiles of $\phi$ (red vertical line) and firing rate (black horizontal line). In all areas, among 25% neurons with the highest firing rates (above the black line), only a few neurons had $\phi$ within the top 25% (to the right of the red line). Most PMd neurons with low-to-moderate firing rates (below the black line) had low $\phi$ (to the left of the red line), likely due to prominent beta-band synchronization in this area. Source data are provided as a Source data file.

that the function $f(v)$ defines the instantaneous firing rate from the subthreshold voltage on single trials and that variability of spike counts for a fixed voltage can be captured with a stochastic spike generation mechanism in our DSR framework. Thus, we applied $f(v)$ to the subthreshold voltage to obtain the instantaneous firing rate, which we then used to map ISIs to the operational time to estimate $\phi$.

We observed that the spiking irregularity estimated from the subthreshold voltage was consistent with our theoretical DSR framework. First, the mean $\hat{\mu}_g$ of the rescaled ISIs was close to 1 ($|\hat{\mu}_g - 1| < 0.031$ for all neurons), validating the condition $\mu_g \approx 1$ s in the DSR framework. This agreement indicates that the subthreshold voltage transformed via $f(v)$ is a good proxy for the instantaneous firing rate in the DSR framework. Second, $\phi$ estimated from the subthreshold voltage corresponded well with $\phi$ estimated from spikes alone by the DSR method (Fig. 3c). The two estimates of $\phi$ differed by only 23% on average. Moreover, our DSR method and the partitioning based on the subthreshold voltage attributed a similar fraction of the total spiking variability (measured by $CV_{raw}^2$ of ISIs in real time) to the spiking irregularity (Fig. 3d, $\phi/CV_{raw}^2$: $0.26 \pm 0.02$ for the DSR method, $0.27 \pm 0.04$ for the subthreshold voltage method, mean $\pm$ std across neurons). These results validate the theoretical assumptions of the DSR framework and confirm the accuracy of the DSR estimation method.

## The diversity of spiking irregularity across neurons and cortical areas

Equipped with the reliable method for partitioning spiking variability, we asked how spiking irregularity $\phi$ varies across neurons and cortical areas. We analyzed spikes recorded from areas spanning different stages of cortical hierarchy: from sensory (visual area V4, 282 neurons[54]), to the association (lateral intraparietal area, LIP, 61 neurons[55]), and premotor regions (dorsal premotor cortex, PMd, 343 neurons[56]), in monkeys performing behavioral tasks (Methods). The spiking irregularity varied systematically across these areas (Fig. 4a). On average, the spiking irregularity was slightly super-Poisson in V4 ($\phi = 1.22 \pm 0.03$, mean $\pm$ std across neurons) and became sub-Poisson and more regular in LIP ($\phi = 0.68 \pm 0.04$) and PMd ($\phi = 0.51 \pm 0.02$). In addition, the diversity of spiking irregularity across neurons within each area also systematically decreased from V4 to LIP to PMd (Fig. 4a, standard deviation of $\phi$ across neurons: 0.45 in V4, 0.34 in LIP, 0.21 in PMd). While nearly all PMd neurons had sub-Poisson spiking irregularity $\phi < 1$, spiking of different V4 neurons ranged from clock-like regular ($\phi \approx 0$) to highly irregular ($\phi > 2$). Since our V4 and PMd recordings similarly sampled all cortical layers, these differences likely reflect differences in the circuitry and functional specialization of neurons in these areas. These results reject

the assumption that the spike generation process is Poisson-like for most cortical neurons[57,58], which is a common assumption in statistical models of neural dynamics on single trials[11,19,21,59,60]. Our results agree with the observation that spiking variability (FF) of many parietal[10,26,27] and PMd neurons[27,28] is sub-Poisson, whereas it is super-Poisson in visual cortical areas[11,26,27,34]. The advance of our results over previous observations is that the spiking irregularity $\phi$ reflects solely the neuron's renewal function and is unaffected by the firing rate fluctuations.

The diversity of $\phi$ across neurons in each area raises a question about the relationship between $\phi$ and the mean firing rate of a neuron. In all areas, neurons with high-firing rates had low spiking irregularity: among the 25% neurons with the highest firing rates, only a few neurons had $\phi$ within the top 25% (Fig. 4b). Low spiking irregularity may allow these neurons to transmit signals with reduced noise to other brain regions[26]. In particular, neurons with high FF and low $\phi$ may transmit high-fidelity information about dynamically changing firing rate on single trials. Overall, the mean firing rate and $\phi$ were negatively correlated in V4 (Pearson correlation coefficient $\rho = -0.27$, $p = 4 \cdot 10^{-6}$, $n = 282$) and LIP ($\rho = -0.34$, $p = 7 \cdot 10^{-3}$, $n = 61$), and not significantly correlated in PMd ($\rho = 0.03$, $p = 0.62$, $n = 343$). This correlation did not arise solely from the refractory period effects of high-firing rate neurons (Supplementary Note 1.5). The lack of correlation in PMd resulted from the prevalence of neurons with low-to-moderate firing rates and low $\phi$, likely due to prominent beta-band synchronization in this area[61,62]. The observed inverse relationship between the mean firing rate and spiking irregularity is nontrivial because $\phi$ in our DSR framework is independent of the firing rate. These findings suggest that spiking irregularity may serve varying functions in different cortical areas, challenging the idea that it stems solely from inherently irreducible sources of noise[63].

## Partial invariance of spiking irregularity

Our finding that $\phi \neq 1$ for most cortical neurons suggests incorporating non-Poisson spiking irregularity into methods for estimating firing rates on single trials. An important consideration for developing such methods is whether the spiking irregularity of a neuron changes dynamically or is approximately constant, invariant to changes in behavioral and cognitive state. The total spiking variability, usually measured by the FF, changes dynamically in many conditions (e.g., during stimulus onset[33] or selective attention[58]), but these changes may reflect modulations of either the firing rate, spiking irregularity[48], or both. Therefore, we proceeded to examine whether $\phi$ of each neuron was invariant or changed across conditions and epochs of behavioral tasks, which correspond to different modes of computation associated with distinct operating regimes of network dynamics.

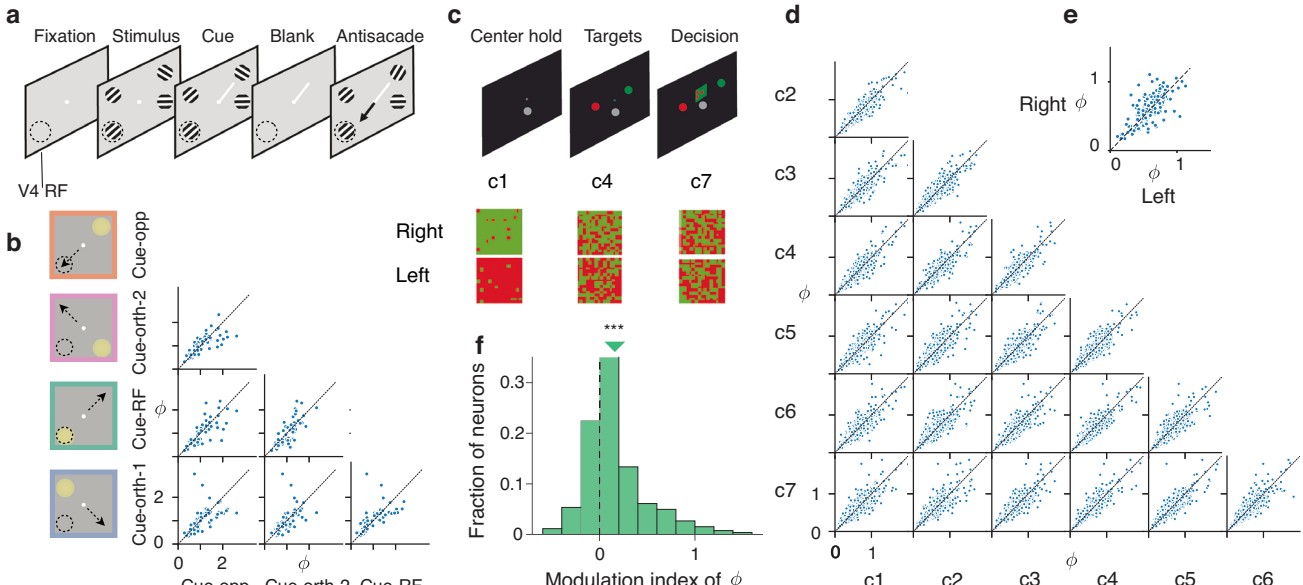

**Fig. 5 | Dependence of spiking irregularity on the behavioral and cognitive state. a** Attention task performed by monkeys during V4 recordings. Monkeys detected an orientation change in one of four peripheral grating stimuli, while an attention cue (short white line) indicated which stimulus was likely to change. Monkeys reported the change with a saccade to the stimulus opposite to the change (black arrow). The cued stimulus was the target of covert attention, while the stimulus opposite to the cue was the target of overt attention. The dashed circle indicates the receptive field location of the recorded neurons (V4 RF). **b** Task conditions (left). On cue-RF trials (green frame), the attended stimulus (yellow circle) was in the RF of recorded neurons. On cue-opposite trials (orange frame), antisaccades were directed to the RF stimulus (black arrow). On cue-orthogonal trials (pink and purple frames), neither the attended stimulus nor the saccade target was in the RF. The scatter plots show estimated $\phi$ for every pair of task conditions. Each dot represents one V4 neuron. **c** Decision-making task performed by monkeys during PMd recordings. Monkeys discriminated the dominant color in a checkerboard stimulus composed of red and green squares and reported their choice by touching the corresponding target (upper panels). Task conditions varied by the response side indicated by the stimulus (left versus right) and stimulus difficulty controlled by seven coherence levels c1 through c7 (lower panels). **d** The scatter plots show estimated $\phi$ for each pair of coherence levels, combined across chosen sides. Each dot is one PMd neuron. **e** The scatter plot shows estimated $\phi$ for right versus left choice trials, combined across coherence levels. Each dot is one PMd neuron. **f** Histogram across PMd neurons of the modulation index of $\phi$ estimated during the fixation (when targets were visible) and decision epochs of the task. The triangle marks the median modulation index, which is significantly greater than zero (***$p = 3 \cdot 10^{-7}$, $n = 262$, one-sided Wilcoxon signed-rank test). Source data are provided as a Source data file.

First, we compared $\phi$ of each V4 neuron across behavioral conditions in the spatial attention task performed by monkeys[15]. In this task, monkeys detected changes in a visual stimulus in the presence of three distractor stimuli and reported the change with an antisaccade response (Fig. 5a). In each trial, a cue indicated the stimulus that was most likely to change, which was thus the target of covert attention, and the stimulus opposite to the cue was the target of overt attention due to the antisaccade preparation. It is well established that variability of neural responses measured by the FF decreases during attention[58,64], but whether the FF decrease results from a reduction in the firing-rate fluctuations, spiking irregularity, or both has not been tested. We compared $\phi$ estimated on trials when the monkeys directed their attention (either covert or overt) to the location of the neuron's receptive field (RF) and on trials when they attended to locations outside the RF and found no significant differences in estimated $\phi$ across these attention conditions (Fig. 5b, $p = 0.10$, $n = 237$, two-sided Friedman test, Methods). This result indicates that spiking irregularity $\phi$ is invariant to the attentional modulation of the network state in V4, and that reduction in FF primarily reflects suppression of the firing-rate fluctuations. Thus, attention stabilizes firing rates over longer timescales without affecting the spiking irregularity of single neurons. This result suggests that attention enhances information transmitted through the firing rates rather than precise spike patterns and provides tight constraints for biophysical network models of attention.

Next, we compared $\phi$ of each PMd neuron across behavioral conditions in the decision-making task performed by monkeys[56]. In this task, monkeys discriminated the dominant color in a static checkerboard stimulus composed of red and green squares and reported their choice by touching the corresponding left or right target (Fig. 5c). The proportion of the same-color squares in the checkerboard (coherence) varied across trials to control the stimulus difficulty, with seven coherence levels used for each left and right response side, resulting in 14 stimulus conditions total. We estimated $\phi$ in each of these conditions separately during the decision epoch of the task after the checkerboard onset and found no significant differences in $\phi$ across stimulus conditions (Fig. 5d, $p = 0.11$, $n = 272$, two-sided Friedman test, Methods). Thus, while PMd dynamics change substantially across conditions in correlation with the chosen side and reaction time[9,56], the spiking irregularity $\phi$ was invariant to these changes. Similarly, spiking irregularity $\phi$ of LIP neurons during the decision epoch was invariant between two-choice and four-choice decision-making tasks ($p = 0.84$, $n = 60$, two-sided Wilcoxon signed-rank test). Thus, in different tasks and cortical areas, we found that the spiking irregularity was invariant across behavioral conditions during the same task epoch.

Finally, we tested whether $\phi$ of a neuron was the same between different task epochs that engage distinct computations. For PMd neurons, we compared $\phi$ during the decision epoch while monkeys were making their choice and during the pre-stimulus fixation period while monkeys held their hand still on a fixation target waiting for the stimulus to appear (Fig. 5c, Methods). The spiking irregularity $\phi$ was significantly greater during the decision epoch than during pre-stimulus fixation (Fig. 5f, modulation index $2(\phi_{decision} - \phi_{fixation})/(\phi_{decision} + \phi_{fixation}) = 0.16$, $p = 0.003$, $n = 262$, two-sided Wilcoxon signed-rank test). The lower spiking irregularity $\phi_{fixation}$ likely reflects elevated beta-band synchronization during the fixation period,

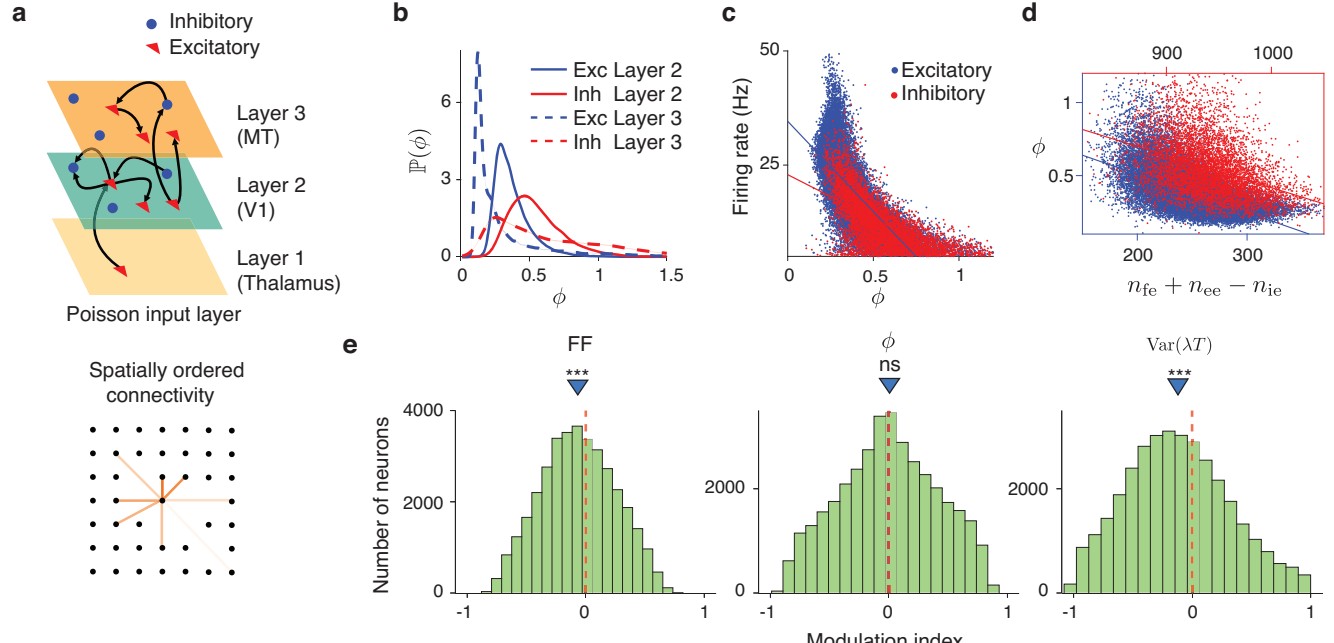

**Fig. 6 | Network mechanisms of diverse spiking irregularity and its invariance during attention. a** The model comprises three layers representing the thalamus, V1, and MT, implemented as spatially organized spiking networks (upper panel). Layer 1 consists of 2500 excitatory Poisson neurons firing at a uniform rate of 10 Hz. Layers 2 and 3 are recurrently coupled balanced networks, each containing 40,000 excitatory and 10,000 inhibitory neurons. Layer 2 receives feedforward excitatory input from Layer 1, and the excitatory neurons in Layer 2 project to Layer 3. Within Layers 2 and 3, neurons form spatially structured recurrent connections, with connection probability decaying with distance according to a Gaussian profile (lower panel; color intensity indicates connection probability). **b** Distributions of spiking irregularity ($\phi$) for excitatory (blue) and inhibitory (red) neurons in Layer 2 (solid lines) and Layer 3 (dashed lines). The diversity of spiking irregularity across Layer 2 neurons aligns with experimental observations (cf. Fig. 4). Excitatory neurons in Layer 3 exhibit highly regular spiking, while inhibitory neurons display a broad range of spiking irregularity. **c** Spiking irregularity of neurons in Layer 2 decreases with the firing rate (blue dots−excitatory neurons, red dots−inhibitory neurons, lines−linear regression). **d** Spiking irregularity of neurons in Layer 2 decreases with the balance in the total number of excitatory and inhibitory connections received by a neuron $n_{fe} + n_{ee} - n_{ie}$. **e** Attentional modulation of FF, spiking irregularity $\phi$, and firing rate variability $Var(\lambda T)$ in Layer 3. The FF modulation index (MI) is defined as $(FF_{attended} - FF_{unattended})/(FF_{attended} + FF_{unattended})$, and similarly for spiking irregularity and firing rate variability. FF is significantly reduced during attention (left, ***$p < 10^{-10}$, $n = 35, 944$, one-sided $t$-test), driven by an attention-mediated reduction in firing rate variability (right, ***$p < 10^{-10}$), while spiking irregularity remains unchanged (center, ns, $p = 0.51$). The dashed red line marks zero, and the blue triangle indicates the mean of the distribution. Source data are provided as a Source data file.

characteristic of the movement preparatory activity in PMd[61,62]. In contrast, the spiking irregularity of LIP neurons was not significantly different between the fixation and decision epochs of the task ($p = 0.84$, $n = 45$, two-sided Wilcoxon signed-rank test). Our results show that the spiking irregularity $\phi$ of a neuron is invariant to alterations of the network state in many conditions, such as different attention states or decisions, but $\phi$ can also change across different modes of network operation associated with distinct computations.

## Network mechanisms of spiking irregularity

We finally asked what biophysical mechanisms could explain our findings that cortical neurons have diverse spiking irregularity, which is inversely related to the mean firing rate and remains nearly constant across many conditions for individual neurons. To test possible mechanisms, we used a spiking recurrent neural network model that accounts for several characteristics of spiking variability in the visual cortex and its attentional modulation[35] (Fig. 6a, Methods). The model consists of a three-layer hierarchy (thalamus-V1-MT), in which the V1 and MT layers are two-dimensional balanced networks of excitatory and inhibitory neurons with spatially ordered recurrent connectivity. The connection probability within recurrent layers falls off with distance between neurons, mimicking lateral connectivity in the visual cortex[65]. The V1 layer receives an excitatory feedforward input from the thalamus layer modeled as Poisson neurons firing at a uniform rate, while excitatory neurons in V1 project to the MT layer. Recurrent interactions in the network generate turbulent dynamics, which give rise to low-dimensional population-wide fluctuations in spiking activity[35].

Since all neurons in the model have the same deterministic voltage threshold for spike generation, the spiking irregularity arises entirely from fluctuating inputs to a neuron shaped by the recurrent network dynamics. Similar to our cortical data, the spiking irregularity $\phi$ in the V1 layer varied broadly across neurons in the network model (Fig. 6b). The diversity of $\phi$ across neurons was not due to heterogeneous single-cell properties, since all excitatory and inhibitory neurons in the model had identical parameters, but due to differences in how neurons were embedded in the network. The connections between neurons in the model are made randomly with distance-dependent probability. Thus, by chance, some neurons receive more excitation than inhibition and reside near or even above the firing threshold and therefore spike regularly, while others receive stronger net inhibition, reside far from the firing threshold, and fire irregularly, driven by large fluctuations. Consistent with this mechanism, $\phi$ was negatively correlated with the average firing rate of a neuron (Fig. 6c) and with the difference between the number of excitatory and inhibitory connections it receives (Fig. 6d). The synaptic input balance is not the sole source of diverse spiking irregularity, as heterogeneous $\phi$ also arises from chaotic dynamics in a balanced random network model, where each neuron receives the same number of excitatory and inhibitory connections[36] (Supplementary Note 1.6, Supplementary Fig. 5). Hence, the spiking irregularity $\phi$ arises from both the recurrent network dynamics and neuron's embedding within this network. These

results show that probabilistic spike generation in doubly stochastic point process models is compatible with a deterministic voltage threshold for spike firing in single neurons.

The distribution of $\phi$ in the spatially ordered balanced network model peaked below one, similar to that for LIP and PMd but not matching the greater diversity and larger values of $\phi$ in V4 (Fig. 6b). This observation suggests that other features of the V4 circuitry not included in this model may contribute to the spiking irregularity, such as a distinct dynamical regime (Supplementary Note 1.6, Supplementary Fig. 5), heterogeneous single-cell properties, cell types, and non-Poisson input. In addition, changes in the network state can dynamically modulate both the average firing rate and spiking irregularity via multiple biophysical mechanisms, such as inputs to excitatory or inhibitory neurons or changes in the membrane conductance (Supplementary Note 1.7, Supplementary Fig. 6), providing a possible mechanism for the change in $\phi$ observed in PMd (Fig. 5f). Thus, measurements of spiking irregularity in experimental data can provide tighter constraints on biophysical neural circuit models.

Finally, it has been shown previously that this spatial network model accounts for the reduction in FF during attention[35], and we tested whether the FF reduction in the model resulted from changes in spiking irregularity or firing rate fluctuations. We model top-down attentional modulation as a static depolarizing input current to MT inhibitory neurons[35] (0.2 mV/ms in the unattended state, 0.4 mV/ms in the attended state, Methods). While FF in the model was significantly reduced during attention (Fig. 6e, FF modulation index $MI_{FF} = -0.06$, $p < 10^{-10}$, $n = 35,944$, two-sided $t$-test), the spiking irregularity remained unchanged (Fig. 6e, $\phi$ modulation index $MI_\phi = -0.001$, $p = 0.51$, $n = 35,944$, two-sided $t$-test), consistent with our experimental observations (Fig. 5b). Accordingly, the source of FF reduction in the model was a decrease in firing rate variability, which we estimated using Eq. (5) (Fig. 6e, $Var(\lambda T)$ modulation index $MI_{Var(\lambda T)} = -0.12$, $p < 10^{-10}$, $n = 35,944$, two-sided $t$-test). Thus, a reduction in FF in the model resulted from a decrease in firing rate fluctuations while spiking irregularity remained unchanged, consistent with our observations in experimental data, therefore supporting the proposed circuit mechanism of attentional modulation. Our method is uniquely suited to detect this dissociation, providing tighter constraints on biophysical circuit models of attention.

## Discussion

We introduced a DSR process, a mathematical framework for partitioning the total spiking variability of neurons into firing rate fluctuations and spiking irregularity. The standard model used to relate dynamically changing firing rates to spikes is an inhomogeneous Poisson process. However, the inhomogeneous Poisson process can only produce a fixed spiking irregularity, corresponding to $\phi = 1$ in our DSR framework. Hence, it cannot account for diverse spiking statistics of neurons, e.g., regular spiking with FF smaller than one. On the other hand, a stationary renewal point process[37] can generate spike trains with a fixed firing rate and any irregularity, from nearly periodic to super-Poisson. A previously proposed non-stationary extension of the renewal process incorporated a time-dependent firing rate but did not account for trial-to-trial firing rate fluctuations[39,66,67]. Our DSR process generalizes these previous models by encompassing both stochastic firing rate fluctuations and a broad spectrum of spiking irregularity. A subset of stationary renewal processes can be expressed as inhomogeneous Poisson processes[68-70], leading to potential ambiguity in assigning variability to the firing rate versus spiking irregularity[29]. We resolve this ambiguity by imposing a minimal set of constraints: the renewal property in the operational time and smoothness of the firing rate over short timescales. With these constraints, our DSR framework enables unambiguous partitioning of variability.

We validated the accuracy of our estimation method on synthetic data with known ground truth. In contrast, we find that previous methods for partitioning spiking variability are less reliable either due to an inability to account for fluctuating firing rate[39] or due to assumptions that do not always hold true[10,28,31]. This latter observation agrees with the previous work showing that any partitioning of variability is ambiguous without an underlying mathematical model[29].

Furthermore, we confirmed that our DSR model aligns with the biophysical properties of neural circuits. This connection had not been tested despite the widespread use of doubly stochastic point processes for modeling spiking activity. In fact, the reliable spiking of cortical neurons in response to time-varying inputs[71] may seem to suggest that stochastic spike generation models are incompatible with circuit biophysics. We show that the spiking irregularity can arise from recurrent dynamics and reflect a neuron's embedding within the network, even when individual neurons have a deterministic voltage threshold for spike generation. Our validation of the DSR model using intracellular voltage recordings and spiking network simulations justifies the widespread use of doubly stochastic models in single-trial spike-train analysis and establishes their connection with the underlying biophysical processes.

We applied our method to survey the spiking irregularity of neurons across sensory, association, and premotor cortical areas. We found that neurons within each area showed a wide range of spiking irregularity, with the greatest diversity in area V4. The diversity of spiking irregularity may arise from several possible sources: heterogeneity in neurons' morphology, cell type, or differences in how a neuron is embedded in the surrounding network. Our simulations of the spiking neural network models show that diverse spiking irregularity can arise from variations in the balance of excitatory and inhibitory inputs across neurons as well as from recurrent network dynamics. We further found that the average spiking irregularity decreased systematically from V4 to LIP to PMd, consistent with previous observations that responses of parietal and PMd neurons are more regular than Poisson[26,27]. These previous studies used various metrics to quantify spiking irregularity in data, but lacked a generative model. Therefore, it is difficult to assess the accuracy of these methods, leaving uncertainty about the reliability of derived conclusions. Our work overcomes these limitations by introducing a mathematical definition of spiking irregularity as a parameter within a generative model, which enables us to verify the accuracy of our estimation method and opens the possibility of integrating spiking irregularity into models of single-trial neural dynamics beyond the standard Poisson assumption. Our results confirm that spiking irregularity decreases along the cortical hierarchy, suggesting it may be related to the functional specialization of cortical areas.

While our results and previous studies[26,27] show that spiking irregularity decreases from visual to association to motor cortical areas, intrinsic neural timescales systematically increase along the cortical hierarchy[72-75]. Intrinsic timescales are defined by the exponential decay rate of the autocorrelation function of spiking activity and typically range from tens to several hundred milliseconds, reflecting primarily slow firing-rate dynamics rather than spiking irregularity. Thus, firing rate timescales and spiking irregularity follow inverse gradients that align with the functional specialization of cortical areas. A precise Bayesian estimation method[76] revealed that spiking activity in the primate visual cortex unfolds on at least two timescales: a fast ~5 ms timescale and a slow ~100 ms timescale[77]. The millisecond range of the fast timescale may partly reflect spiking irregularity. Moreover, the slow—but not the fast—timescale increased during selective attention[77], consistent with our observation that spiking irregularity remains invariant while firing rate fluctuations are stabilized during attention. Together, these findings suggest that spiking irregularity and intrinsic timescales reflect distinct yet complementary features of neural dynamics, each influencing how cortical areas process information over time.

We tested whether spiking irregularity is a neuron-specific constant, invariant to changes in network dynamics due to variations in behavioral

and cognitive state. Indeed, the spiking irregularity was invariant in many cases, such as across different attention states or decision difficulties. However, we also found that the spiking irregularity of PMd neurons changed between different epochs of the task[48], which indicates that spiking irregularity can also change as a function of the network state. Our spiking network simulations show that depolarization of excitatory neurons combined with an increase in their membrane conductance can simultaneously increase the spiking irregularity and firing rate, similar to the modulation we observed in PMd. The effective membrane conductance increases with depolarization in conductance-based models of spiking neurons[78,79], which are therefore especially useful for studying mechanisms modulating spiking irregularity.

Changes in neural variability can furnish insights into mechanisms of diverse brain functions[10,11,80]. For example, FF decreases in many brain regions during sensory stimulus onset[33], motor preparation[3,81], and selective attention[58,82], which has been interpreted as a mechanism for enhancing the fidelity of neural representations. However, FF mixes contributions from both firing rate fluctuations and spiking irregularity, making such a mechanistic interpretation more challenging[48]. Going beyond FF, some studies partitioned neural variability into firing rate fluctuations and spiking irregularity[10,31,48]. This partitioning approach revealed that firing rate variability on longer timescales increases through the decision period, while assuming that spiking irregularity is fixed across different task epochs[10]. In contrast, we found that although spiking irregularity was invariant in many cases, it changed across different epochs of the decision-making task in PMd. Thus, our DSR framework and estimation method enable more nuanced analyses of neural variability, opening a possibility to identify how changes in the firing rate variability and spiking irregularity independently contribute to neural computations.

While our DSR model assumes that ISIs are independent in operational time, it can generate serial ISI correlations in real time through temporally correlated instantaneous firing rates. Such serial correlations between ISIs are commonly observed in data[83,84] and can arise from either correlated input or firing rate adaptation in mechanistic integrate-and-fire models[85–88]. Our DSR model can incorporate ISI correlations in real time via temporally correlated firing rate fluctuations, while maintaining approximately uncorrelated ISIs in operational time (Supplementary Note 1.8). Thus, our DSR framework can be broadly applied to model spiking responses.

While an inhomogeneous Poisson process is widely used in methods for inferring latent neural dynamics on single trials[18–25], our results highlight the need for incorporating non-Poisson spiking irregularity into these methods. One approach for incorporating non-Poisson spiking is to augment the inhomogeneous Poisson process with the instantaneous firing rate that depends on the spike history[60,89,90]. Our DSR framework offers two additional approaches for including non-Poisson spiking irregularity into latent variable models. First, we can estimate $\phi$ from data with our DSR method and then model the spike generation from the firing rate as a renewal point process with $g(\cdot)$ being a gamma distribution uniquely defined by the estimated $\phi$. Second, we can simultaneously infer the distribution $g(\cdot)$ and latent dynamics[91–93]. In our framework, the inference of $g(\cdot)$ amounts to the estimation of a single parameter $\phi$ per neuron and thus is maximally parameter-efficient. Finally, our DSR framework provides an accurate metric for evaluating the goodness of fit of latent variable models, whereas Poisson likelihood may produce misleading results when applied to spikes with non-Poisson statistics[30].

Together, our results uncover the great diversity in the spiking irregularity across neurons, cortical areas, and cognitive states, which cannot be captured with the conventional inhomogeneous Poisson model. Our theoretical framework and estimation method provide a flexible tool for quantifying spiking variability to investigate its role in neural computation and the underlying biophysical mechanisms. Our DSR point process provides a flexible model to capture the broad

spectrum of spiking irregularity of cortical neurons and improve the precision of methods for estimating latent dynamics on single trials.

## Methods
### Partitioning variability

For a pair of random variables $X$ and $Y$, the law of total variance (LOTV) decomposes the variance of $Y$ into two parts: $\mathrm{Var}(Y) = \mathrm{E}[\mathrm{Var}(Y|X)] + \mathrm{Var}(\mathrm{E}[Y|X])$. We assume that $\lambda(t)$ is approximately constant within a time bin of size $T$, then choosing $X = \lambda$ and $Y = N_T$, we obtain Eq. (3): $\mathrm{Var}(N_T) = \mathrm{E}[\mathrm{Var}(N_T|\lambda)] + \mathrm{Var}(\mathrm{E}[N_T|\lambda])$. Next, we calculate $\mathrm{E}[N_T|\lambda]$ and $\mathrm{Var}(N_T|\lambda)$ for a DSR point process $\{g(\cdot), \lambda(t)\}$.

First, we consider a simple renewal point process fully defined by its ISI probability density $f(x)$, meaning that after generating a spike, the probability of the next spike occurring within the interval $[x, x + dx]$ is $f(x)dx$. Denoting the first three central moments of $f(x)$ by $\mu$, $\sigma^2$, and $\mu_3$, we show that the mean and variance of the spike count $N_T$ in a bin with the size $T$ are (Theorem 1 in Supplementary Note 1.1):

$$E(N_T) = \frac{T}{\mu}, \tag{9}$$

$$\mathrm{Var}(N_T) = \frac{\sigma^2}{\mu^3}T + \frac{\sigma^4}{2\mu^4} + \frac{1}{6} - \frac{\mu_3}{3\mu^3} + \mathcal{O}(T^{-1}). \tag{10}$$

Next, we consider a DSR point process $\{g(\cdot), \lambda(t)\}$. Assuming $\lambda(t)$ changes on a timescale longer than $T$, we can consider $\lambda$ to be approximately constant within a bin. Then, within a single bin, the spike-generating process is a renewal point process fully defined by its ISI probability density, which we denote by $f_\lambda$ since it depends on the value of $\lambda$ in the bin. We derive that $\mathrm{E}[N_T|\lambda] = \lambda T$ (Theorem 2 in Supplementary Note 1.1), which we substitute in Eq. (3) to get $\mathrm{Var}(N_T) = \mathrm{E}[\mathrm{Var}(N_T|\lambda)] + \mathrm{Var}(\lambda T)$. We further derive that for a moderately large bin size $T > 1/\mathrm{E}[\lambda]$, we can express $\mathrm{E}[\mathrm{Var}(N_T|\lambda)]$ via the first three moments of the probability density $g(\cdot)$ as stated in Eq. (4) (Theorem 4 in Supplementary Note 1.1). With these results, we obtain

$$\mathrm{Var}(N_T) = \mathrm{Var}(\lambda T) + \left(\frac{\sigma_g}{\mu_g}\right)^2 \mathrm{E}[N_T] + \frac{1}{6} + \frac{1}{2}\left(\frac{\sigma_g}{\mu_g}\right)^4 - \frac{1}{3}\frac{\mu_{3g}}{\mu_g^3} + \mathcal{O}(T^{-1}). \tag{11}$$

To simplify the partitioning equation, we assume that $g(\cdot)$ belongs to the two-parameter family of continuous probability distributions and is uniquely determined by its first two moments $\mu_g$ and $\sigma_g^2$. Since $\mu_g = 1\,\mathrm{s}$, the probability density $g(\cdot)$ is uniquely determined by $\phi = \sigma_g^2/\mu_g^2$. Thus, the third central moment is a function of $\phi$, which we denote by $\mu_3 = \psi(\phi)$. With this parametrization, we obtain our general partitioning equation:

$$\mathrm{Var}(N_T) = \mathrm{Var}(\lambda T) + \phi\, \mathrm{E}[N_T] + \frac{1}{6} + \frac{1}{2}\phi^2 - \frac{1}{3}\psi(\phi) + \mathcal{O}(T^{-1}). \tag{12}$$

As a special case of the two-parameter family, we consider $g(\cdot)$ to be a gamma distribution[38,42,43]

$$g(\tau) = \frac{1}{\Gamma(k)\theta^k}\tau^{k-1}e^{-\frac{\tau}{\theta}}, \tag{13}$$

which we can reparameterize in terms of $\phi$:

$$g(\tau) = (\Gamma(\phi^{-1}))^{-1}\phi^{-\frac{1}{\phi}}\tau^{\frac{1-\phi}{\phi}}e^{-\frac{\tau}{\phi}}. \tag{14}$$

In the special case of $\phi = 1$, the ISI distribution reduces to the exponential distribution $g(\tau) = e^{-\tau}$, which corresponds to the Poisson spiking process. We can compute the central moments of the gamma

distribution $g(\tau)$ in terms of $\phi$, specifically, $\mu_g = k\theta = 1$, $\sigma_g^2 = k\theta^2 = \phi$, and $\mu_{3g} = 2k\theta^3 = 2\phi^2$. Substituting these expressions in Eq. (11), we obtain our final partitioning equation Eq. (5).

## Estimation methods for $\phi$

We develop a DSR method for estimating $\phi$ from spike data. The data consist of spike times for multiple trials. We choose a bin size $T$ and estimate $E[N_T]$ and $\mathrm{Var}(N_T)$ as the mean and variance of spike counts across trials in bins $[t, t + T]$, where $t$ is a time within a trial. Besides $\phi$, the only other unknown term in the partitioning equation Eq. (12) is $\mathrm{Var}(\lambda T)$. Using the fact that $\mathrm{Var}(\lambda T) = T^2 \mathrm{Var}(\lambda)$ for every bin, we express this unknown part through other terms

$$T^2 \,\mathrm{Var}\,(\lambda) \approx \mathrm{Var}\,(N_T) - \phi\,E[N_T] - \frac{1}{6} - \frac{1}{2}\phi^2 + \frac{1}{3}\psi(\phi). \qquad (15)$$

In this equation, $E[N_T]$, $\mathrm{Var}(N_T)$, and $T^2\mathrm{Var}(\lambda)$ are all functions of the bin size $T$. Thus, by changing the bin size, we can obtain a system of two equations for two bins $[t, t + T]$ and $[t, t + \widetilde{T}]$:

$$\begin{cases} T^2 \,\mathrm{Var}\,(\lambda) \approx \mathrm{Var}\,(N_T) - \phi\,E[N_T] - \frac{1}{6} - \frac{1}{2}\phi^2 + \frac{1}{3}\psi(\phi), \\ \widetilde{T}^2 \,\mathrm{Var}\,(\lambda) \approx \mathrm{Var}\,(N_{\widetilde{T}}) - \phi\,E[N_{\widetilde{T}}] - \frac{1}{6} - \frac{1}{2}\phi^2 + \frac{1}{3}\psi(\phi). \end{cases} \qquad (16)$$

Denoting $\alpha = \widetilde{T}/T > 1$, we eliminate the unknown $\mathrm{Var}(\lambda)$ in this system of equations and obtain a single equation in which $\phi$ is the only unknown variable:

$$(\alpha^2 - 1)\left(\frac{\psi(\phi_{\mathrm{DSR}})}{3} - \frac{\phi_{\mathrm{DSR}}^2}{2}\right) - (\alpha^2\,E[N_T] - E[N_{\widetilde{T}}])\phi_{\mathrm{DSR}}$$
$$+ \alpha^2 \,\mathrm{Var}\,(N_T) - \mathrm{Var}\,(N_{\widetilde{T}}) - \frac{\alpha^2 - 1}{6} = 0. \qquad (17)$$

Here we denote the solution of this equation by $\phi_{\mathrm{DSR}}$, which is an estimate of $\phi$ with the DSR method. The coefficients in this equation include four terms $E[N_T]$, $E[N_{\widetilde{T}}]$, $\mathrm{Var}(N_T)$ and $\mathrm{Var}(N_{\widetilde{T}})$ that can be directly estimated from data with a moderate number of trials. In the case when $g(\cdot)$ is the gamma distribution, $\mu_{3g} = \psi(\phi) = 2\phi^2$, and we obtain a quadratic equation for $\phi_{\mathrm{DSR}}$:

$$\frac{\alpha^2 - 1}{6}\phi_{\mathrm{DSR}}^2 - (\alpha^2\,E[N_T] - E[N_{\widetilde{T}}])\phi_{\mathrm{DSR}} + \alpha^2\,\mathrm{Var}\,(N_T)$$
$$- \mathrm{Var}\,(N_{\widetilde{T}}) - \frac{\alpha^2 - 1}{6} = 0. \qquad (18)$$

To derive this equation, we assumed that for every realization of $\lambda(t)$, the firing rate $\lambda(t)$ changes slowly relative to the timescales $T$ and $\widetilde{T} = \alpha T$. Thus, we assume that $\lambda(t)$ is constant within the bin $\widetilde{T}$, hence, $\alpha$ cannot be too large. However, if $\alpha$ is very close to 1, the two equations for different bin sizes $T$ and $\widetilde{T}$ are nearly identical, leading to a large error in estimated $\phi$. Through simulations, we found that $\alpha = 2$ produces accurate results for wide ranges of parameters, and therefore, we set $\alpha = 2$ in all analyses. Substituting $\alpha = 2$ in Eq. (18), we obtain the final estimation equation Eq. (6).

We compared the accuracy of our estimation method with two previously proposed methods. The first method, the DTR[39], assumes that the time-dependent firing rate is the same on every trial, deterministically locked to a trial event. The method then estimates the firing rate as the average spike count in a bin across trials: $\hat{\lambda}(t_i) = 1/K \sum_{k=1}^{K} N_{T,i}^k$, where $N_{T,i}^k$ is the number of spikes in the bin $[t_i - \frac{T}{2}, t_i + \frac{T}{2}]$ in the $k$th trial, and $K$ is the number of trials. If the ground-truth firing rate is the same on each trial, $\phi_{\mathrm{DTR}}$ converges to the ground-truth $\phi$ for large trial number: $\lim_{\mathrm{Var}[\lambda(t)]\to 0, K\to\infty} \phi_{\mathrm{DTR}} = \phi$ (Supplementary Note 1.3). However, in the presence of trial-to-trial firing rate fluctuations, we derive that the estimation error of this

method is $\phi_{\mathrm{DTR}} - \phi = (A - 1)(\phi + 1)$, where $A$ is a monotonically increasing function of the average firing rate variance (Supplementary Note 1.3).

The second family of methods[10,31,47] assumes that the point process variance is proportional to the mean spike count $E[\mathrm{Var}(N_T|\lambda)] = \phi E[N_T]$, where $\phi$ is a neuron-specific constant (Eq. 7). This assumption was motivated by the renewal theory[37], which states that for a stationary renewal process with a constant firing rate, the FF converges to the squared coefficient of variation of ISIs ($\mathrm{CV}^2$) in the limit of an infinite bin size: $\lim_{T\to\infty} \mathrm{FF} = \mathrm{CV}^2$. However, this relation strictly applies only in the limit of infinite bin size ($T \to \infty$) and a constant firing rate[94], and for a finite bin size, the FF of a renewal process depends on both the bin size and firing rate. Using Eq. (4), we derive that for a renewal process with a constant firing rate in a finite bin size, it holds $\mathrm{FF} = \mathrm{CV}^2 + \frac{c_1}{E(N_T)} + \frac{1}{T} \cdot \frac{c_2}{E(N_T)} + \mathcal{O}(T^{-2})$, where the coefficients $c_1$ and $c_2$ depend on $g(\cdot)$. This relation shows that $\phi$, defined via Eq. (7) with a finite bin size, is not a constant characterizing the renewal process, but a function of the bin size and firing rate. Therefore, any method for estimating $\phi$ using Eq. (7) with a finite bin size will produce results that do not uniquely characterize the spiking irregularity of a neuron, but depend on nuisance parameters such as the bin size and firing rate. While it is possible to accurately estimate $\phi$ (i.e., $\mathrm{CV}^2$ of ISIs of a renewal process) by considering the asymptotic behavior of $\mathrm{FF}(T)$ for large bin sizes[48], this asymptotic method requires the firing rate to be constant over time bins $T$ longer than fast timescales involved in behavior, e.g., decision-making.

Based on Eq. (7), the MR method[10] estimates $\phi$ as the minimum FF across all time bins. We show that for DSR processes, the estimation error of this method is (Supplementary Note 1.4)

$$\phi_{\mathrm{MR}} - \phi \approx \min_t\left\{T\frac{\mathrm{Var}\,(\lambda(t))}{E[\lambda(t)]} + \frac{1}{T} \cdot \frac{\frac{1}{6} + \frac{1}{2}(\phi)^2 - \frac{1}{3}\psi(\phi)}{E[\lambda(t)]}\right\}. \qquad (19)$$

Other methods were also proposed for estimating $\phi$ in Eq. (7) with better accuracy than the MR method under the assumption that the firing rate obeys an unbounded drift-diffusion process[31,47]. However, these methods require a priori knowledge of the firing rate dynamics. In addition, they start from the same premise, Eq. (7), as the MR method, and therefore have similar limitations for a finite bin size.

## Synthetic data generation

For the drift-diffusion model with sticky boundaries (Figs. 1 and 2), we generate the firing rate from

$$\frac{d\lambda(t)}{dt} = \begin{cases} 0, & \lambda(t) = b_l, \\ \nu + \sqrt{2D}\xi(t), & b_l < \lambda(t) < b_u, \\ 0, & \lambda(t) = b_u. \end{cases} \qquad (20)$$

Here, $\nu$ is the drift, $\xi(t)$ is a white Gaussian noise $\langle\xi(t)\rangle = 0$, $\langle\xi(t)\xi(t')\rangle = \delta(t - t')$, $D$ is the diffusion coefficient, and $b_l$ and $b_u$ are the lower and upper boundary values, respectively. We use $b_l = 1\,\mathrm{Hz}$, $b_u = 20\,\mathrm{Hz}$, $\nu = 0\,\mathrm{Hz} \cdot \mathrm{ms}^{-1}$, and the initial firing rate value $\lambda = 10\,\mathrm{Hz}$ (Fig. 1) and $b_l = 1\,\mathrm{Hz}$, $b_u = 60\,\mathrm{Hz}$, $\nu = 0.0138\,\mathrm{Hz} \cdot \mathrm{ms}^{-1}$, and the initial firing rate value $\lambda = 30\,\mathrm{Hz}$ (Fig. 2).

## Estimation of $\phi$ from spike data

To estimate $\phi$ with the DSR method in synthetic and experimental data, we found $\phi$ as a solution of Eq. (6) for each time point $t_i$ within an analysis window with two time bins $[t_i, t_i + T]$ and $[t_i, t_i + 2T]$. We then obtained the final $\phi$ by averaging the results across all time points $t_i$ within the analysis window. To ensure that Eq. (5) holds, we require $T > 1/E[\lambda]$. Since we assume the firing rate is constant within a bin, we also need to choose a bin size as small as possible. To satisfy both conditions, we set $T = 2/E[\lambda]$ for each neuron in experimental and

synthetic data, where $E[\lambda]$ is the average firing rate of the neuron over the analysis period. We confirmed that our inference method is robust and largely insensitive to bin size across a wide range of values, and remains reliable even in the presence of occasional rapid changes in firing rate (Supplementary Note 1.2).

To estimate $\phi$ with the DTR method (Fig. 2), we estimated the trial-averaged firing rate using a 60 ms sliding window with 10 ms increments. We used this firing rate and Eq. (1) to map spike times from real time $t$ to operational time $t'$. We then computed $\phi$ as the squared coefficient of variation $CV^2$ of ISIs in operational time. For the MR method (Fig. 2), we used Eq. (8) with a bin size of 60 ms.

### Intracellular voltage recordings

We used a previously described dataset[32] of whole-cell intracellular recordings of membrane potential from PV neurons in L2/3 of barrel cortex in awake head-restrained mice (six 5–10 week old female and male PV-IRES-Cre mice). All experiments were carried out in accordance with protocols approved by the Swiss Federal Veterinary Office (authorization VD1628). The sampling rate of the membrane potential measurements was 20 KHz. We only analyzed neurons with a trial-averaged firing rate of at least 20 Hz that had data for at least 10 trials of 20 s duration each. These criteria yielded eight neurons from five female mice.

We estimated the function that relates subthreshold membrane potential to instantaneous firing rate, similar to previous studies[52,53]. First, we removed action potentials from the voltage traces. We define the spike time as the time when the membrane potential crosses a −30 mV threshold (the average membrane potential for cells is typically much lower at about −60 mV). We then remove the segment of the voltage trace from 3 ms before to 5 ms after each spike time and linearly interpolate between these two points. Second, we segment the recorded voltage trace in $\Delta t = 50$ ms time bins. For each time bin $i$, we compute the average membrane potential $V_i$ and the number of spikes $N_i$ within that time bin. We plot $N_i/\Delta t$ versus $V_i$ for all time bins (blue dots in Fig. 3b). Next, we divide the voltage range [−68, −40] mV into $\Delta V = 1$ mV bins $\Delta V_k$ ($k = 1, 2, \cdots, 28$). We find the set of all data points falling within the $k$th bin $S_k = [\ \forall i : V_i \in \Delta V_k]$, and compute the average firing rate

$$r_k = \frac{1}{|S_k|\Delta t} \sum_{i \in S_k} N_i \tag{21}$$

and the average voltage $V_k = \frac{1}{2} \cdot (-68 + k\Delta V)$ in that bin. Finally, we fit this average relationship with a spline to approximate the dependence of the instantaneous firing rate $r_k$ on subthreshold membrane potential $V_k$ using the deterministic function $f(v)$.

### Neural recording data

We analyzed three experimental datasets described previously: recordings from area V4 during a spatial selective attention task[54], recordings from the LIP during two-choice and four-choice decision-making tasks[55], and recordings from the dorsal premotor cortex (PMd) during a decision-making task[56]. Experimental procedures for the V4 and PMd datasets were in accordance with the NIH Guide for the Care and Use of Laboratory Animals, the Society for Neuroscience Guidelines and Policies, and the Stanford Institutional Animal Care and Use Committee. Experimental procedures for the LIP dataset were in accordance with the NIH Guide for the Care and Use of Laboratory Animals and approved by the University of Washington Animal Care Committee.

**V4 dataset.** During recordings, the monkeys (G and B, *Macaca mulatta*, male, between 6 and 9 years old) detected orientation changes in one of the four peripheral grating stimuli while maintaining central fixation. Each trial started by fixating a central fixation dot on the screen, and after several hundred milliseconds (170 ms for monkey B and 333 ms for monkey G), four peripheral stimuli appeared. Following a 200–500 ms period, a central attention cue indicated the

stimulus that was likely to change with ~90% validity. The cue was a short line from a fixation dot pointing toward one of the four stimuli, randomly chosen on each trial with equal probability. After a variable interval (600–2200 ms), all four stimuli disappeared for a brief moment and reappeared. Monkeys were rewarded for correctly reporting the change in orientation of one of the stimuli (50% of trials) with an antisaccade to the location opposite to the change, or maintaining fixation if none of the orientations changed. Due to the anticipation of an antisaccade response, the cued stimulus was the target of covert attention, while the stimulus in a location opposite to the cue was the target of overt attention. In the cue-RF condition (Cue-RF), the cue pointed to the stimulus in the RFs of the recorded neurons (covert attention). In the cue-opposite condition (Cue-opp), the cue pointed to the stimulus opposite to the RFs (overt attention). The remaining two cue directions were cue-orthogonal conditions (Cue-orth-1 and Cue-orth-2), in which monkeys attended away from the RFs.

Recordings were performed in the visual area V4 with linear array microelectrodes inserted perpendicularly to the cortical layers. Data were amplified and recorded using the Omniplex system (Plexon). Arrays were placed such that the RFs of recorded neurons largely overlapped. Each array had 16 channels with 150 µm center-to-center spacing. After spike sorting and quality control, the dataset had 285 well-isolated single neurons from two monkeys.

**LIP dataset.** During recordings, two monkeys (*Macaca mulatta*, male, between 11 and 13 years old) performed the random dot motion discrimination task with either two or four choice alternatives. After a variable fixation period, two or four peripheral choice targets appeared to signal the direction alternatives on the trial. After a random delay (250–800 ms), dynamic random dot motion was displayed around the fixation point. The percentage of coherently moving dots on each trial controlled the task difficulty. Monkeys reported the net direction of motion in dynamic random dots by making a saccade to a peripheral choice target. The motion stimulus ended by the time of saccade initiation.

Single neuron activity was recorded with Alpha Omega electrodes introduced into the LIP area. Data were amplified and recorded using the Omniplex system (Plexon). Neurons were selected according to anatomical and physiological criteria, and all had spatially selective responses during the delay on the overlap and memory saccade tasks[55]. The dataset consists of extracellular recordings from 70 well-isolated neurons.

**PMd dataset.** During recordings, the monkeys (T and O, *Macaca mulatta*, male, between 6 and 9 years old) discriminated the dominant color in a static checkerboard stimulus composed of red and green squares and reported their choice by touching the corresponding target. At the start of each trial, a monkey touched a central target and fixated on a cross above the central target. After a short holding period (300–485 ms), red and green targets appeared on the left and right sides of the screen. The colors of each side were randomized on each trial. After another short delay (400–1000 ms), the checkerboard stimulus appeared on the screen at the fixation cross, and the monkey had to move its hand to the target matching the dominant color in the checkerboard. The difficulty of the task was parameterized by an unsigned stimulus coherence expressed as the absolute difference between the number of red ($R$) and green ($G$) squares, normalized by the total number of squares $|R-G|/(R+G)$. The checkerboard was $15 \times 15$ squares, which led to a total of 225 squares. The task was performed with 7 different unsigned coherence levels for monkey T and 8 levels for monkey O, and we analyzed the 7 overlapping coherence levels for two monkeys. Since PMd neurons are selective for the chosen side but not for color[56], we divided the trials according to the side indicated by the stimulus (left or right) for each coherence level, resulting in 14 analyzed conditions in total.

Neural activity was recorded with a linear multi-contact electrode (U-probe) with 16 channels with 150 µm center-to-center spacing. After spike sorting and quality control, the dataset had 801 well-isolated single neurons from two monkeys.

## Selection of units for the analyses

For all datasets, we selected units for our analyses based on two criteria: (i) we included conditions that had at least 20 trials, (ii) we included units that had at least 500 spikes in total across all trials of each condition within the analysis window.

For the V4 dataset, we estimated $\phi$ during the attention epoch of the trial in a window from 400 to 2300 ms aligned to the attention cue onset. 282 out of 285 single units passed the two criteria for all attention conditions combined and were used for quantifying the diversity of $\phi$ across neurons. 237 single units passed the two criteria in each of the four attention conditions and were used for comparing $\phi$ across attention conditions.

For the LIP dataset, we estimated $\phi$ in two trial epochs: fixation epoch, a window from −200 to 0 ms aligned to the stimulus onset; and decision epoch, a window from 0 to the minimum of 1200 ms or reaction time aligned to the stimulus onset. Sixty-one out of 70 single units passed the two criteria during the decision epoch and were used to quantify the diversity of $\phi$ across neurons. 60 units passed the two criteria for both two-choice and four-choice decision-making tasks and were used for comparing $\phi$ across the tasks. Forty-five units passed the two criteria for both the fixation and decision epochs and were used for comparing $\phi$ across trial epochs.

For the PMd dataset, we estimated $\phi$ in two trial epochs: fixation epoch, a window from −600 to 0 ms aligned to the stimulus onset; and decision epoch, a window from 0 to the minimum of 500 ms or reaction time aligned to the stimulus onset. 343 out of 801 single units passed the two criteria during the decision epoch and were used to quantify the diversity of $\phi$ across neurons. Two hundred seventy-two units passed the two criteria for each of 14 different task conditions (2 response sides times 7 stimulus difficulties) and were used for comparing $\phi$ across task conditions. Two hundred sixty-two units passed the two criteria for both the fixation and decision epochs and were used for comparing $\phi$ across two epochs.

## Comparing spiking irregularity across task conditions

To test whether $\phi$ is a neuron-specific constant invariant to changes in behavioral and cognitive states, we estimated $\phi$ of each neuron separately in each task condition. We then used Demšar's comparison test[95] on these populations of paired measurements of $\phi$ using the autorank package[96]. The test is conducted for $M$ populations ($M$ is the number of compared conditions) with $N$ paired samples ($N$ is the number of neurons). The family-wise significance level of the tests is $\alpha = 0.05$. First, we test the null hypothesis that the population is normal for each population. This null hypothesis was rejected for at least some populations in all our tests (detailed summary of the results in Supplementary Note 1.9). Since we have more than two populations and some of them are not normal, we use the non-parametric Friedman test as an omnibus test to determine if there are any significant differences between the median values of the populations. We use the post-hoc Nemenyi test to infer which differences are significant. We report the median, the median absolute deviation, and the mean rank among all populations over the samples (Supplementary Tables 1–5). Differences between populations are significant if the difference of the mean ranks is greater than the critical distance CD of the Nemenyi test.

For comparison between attention conditions in V4 data, we estimated $\phi$ separately on correct trials of each attention condition during the cue period (from 400 to 2300 ms aligned to cue onset), combined across different stimulus orientations. For comparison between two-choice and four-choice decision tasks in LIP data, we estimated $\phi$ separately for each task during the decision epoch (from 0 ms to the minimum of 1200 ms or reaction time aligned to stimulus onset), combined across different coherence levels and chosen sides. For comparison between fixation and decision epochs in LIP data, we estimated $\phi$ separately during the fixation (from −200 to 0 ms aligned to stimulus onset) and decision epochs, combined across two-choice and four-choice decision tasks. For comparison between coherence levels in PMd data, we estimated $\phi$ separately for each coherence level during the decision epoch (from 0 ms to the minimum of 500 ms or reaction time aligned to stimulus onset), combined across the left and right chosen sides. For comparison between left and right choices in PMd data, we estimated $\phi$ separately on left and right choice trials during the decision epoch, combined across coherence levels. For comparison between fixation and decision epochs in PMd data, we estimated $\phi$ separately during the fixation (from −600 to 0 ms aligned to stimulus onset) and decision epochs, combined across coherence levels and chosen sides.

The Friedman test failed to reject the null hypothesis that there is no difference in the central tendency of the populations for comparison between attention conditions in V4 data ($p = 0.10$, $M = 4$, $N = 237$, Supplementary Table 1), two-choice and four-choice decision tasks in LIP data ($p = 0.109$, $M = 2$, $N = 60$, Supplementary Table 2), fixation and decision epochs in LIP data ($p = 0.84$, $M = 2$, $N = 45$, Supplementary Table 3), coherence levels and left and right choices in PMd data ($p = 0.11$, $M = 14$, $N = 272$, Supplementary Table 4). Wilcoxon's signed-rank test rejected the null hypothesis that there is no difference in the central tendency of the populations for comparison between fixation and decision epochs in PMd data ($p = 3.35 \cdot 10^{-12}$, $M = 2$, $N = 262$, Supplementary Table 5).

## Spiking neural network model

We simulated the three-layer spatial balanced spiking network model using the parameters and code from ref. 35. The network consists of three layers. Layer 1 contains $N_f = 2500$ excitatory neurons that generate spikes as independent Poisson processes at a uniform rate $f_{in} = 10$ Hz. Layers 2 and 3 are recurrently connected networks, each comprising $N_e = 40{,}000$ excitatory ($\alpha = e$) and $N_i = 10{,}000$ inhibitory neurons ($\alpha = i$). All neurons ($N_f$, $N_e$, and $N_i$) are uniformly distributed on a unit square. Layer 1 provides feedforward input to Layer 2, where each neuron in Layer 1 connects to exactly $P_{fe}^{(2)} \cdot N_e$ excitatory and $P_{fi}^{(2)} \cdot N_i$ inhibitory neurons in Layer 2, with connection probabilities $P_{fe}^{(2)} = 0.1$ and $P_{fi}^{(2)} = 0.05$. Only excitatory neurons in Layer 2 project to Layer 3, where each neuron connects to exactly $P_{fe}^{(3)} \cdot N_e$ excitatory and $P_{fi}^{(3)} \cdot N_i$ inhibitory neurons, with $P_{fe}^{(3)} = 0.05$ and $P_{fi}^{(3)} = 0.05$. Both Layer 2 and Layer 3 are recurrently connected. Within each layer, every excitatory neuron connects to exactly $P_{ee} \cdot N_e$ excitatory and $P_{ei} \cdot N_i$ inhibitory neurons, while every inhibitory neuron connects to exactly $P_{ie} \cdot N_e$ excitatory and $P_{ii} \cdot N_i$ inhibitory neurons. The corresponding connection probabilities are $P_{ee} = 0.01$, $P_{ii} = 0.04$, $P_{ei} = 0.03$, and $P_{ie} = 0.04$.

For each population, neurons are uniformly distributed on a unit square where the position $(x_j, y_j)$ of neuron $j$ ($1 \leqslant j \leqslant N$) is $0 \leqslant x_j = \frac{1}{\sqrt{N}-1} \cdot \mathrm{mod}(j-1, N)$, $y_j = \frac{1}{\sqrt{N}-1} \cdot \mathrm{floordivision}(j-1, N) \leqslant 1$, where $N$ is the total number of neurons in that population. The probability of a synaptic connection depends on the distance between units. Unit $i$ from group $\alpha \in \{f, e, i\}$ at location $(x_i, y_i)$ connects to unit $j$ from group $\beta \in \{f, e, i\}$ at location $(x_j, y_j)$ with the probability $\mathbb{P}_\alpha(i, j) = f(x_j - x_i, \sigma_\alpha) f(y_j - y_i, \sigma_\alpha)$, where

$$f(r, \sigma_\alpha) = \frac{1}{\sigma_\alpha \sqrt{2\pi}} \sum_{k=-\infty}^{\infty} \exp\left[-\frac{(r+2k)^2}{2\sigma_\alpha^2}\right], \qquad (22)$$

**Table 1 | Single neuron parameters in the spiking network model**

| Type | $\tau_m = \frac{C_m}{g}$ | $E_L$ | $V_T$ | $V_{th}$ | $\Delta_T$ | $V_{re}$ | $\tau_{ref}$ | $\tau_d$ | $\tau_r$ |
|---|---|---|---|---|---|---|---|---|---|
| Excitatory | 25 ms | −60 mV | −50 mV | −10 mV | 2 mV | −65 mV | 1.5 ms | 5 ms | 1 ms |
| Inhibitory | 10 ms | −60 mV | −50 mV | −10 mV | 0.5 mV | −65 mV | 0.5 mv | 8 ms | 1 ms |

**Table 2 | Connectivity parameters in the spiking network model**

| | | | |
|---|---|---|---|
| $J^{(1)fe} = 140$ mV | $J^{(2)fe} = 25$ mV | $J^{ee} = 80$ mV | $J^{ie} = -240$ mV |
| $J^{(1)fi} = 100$ mV | $J^{(2)fi} = 15$ mV | $J^{ei} = 40$ mV | $J^{ii} = -300$ mV |

with $-1 \leqslant r \leqslant 1$, $\sigma_f^{(1)} = 0.05$, $\sigma_e^{(2)} = \sigma_i^{(2)} = 0.1$, $\sigma_f^{(2)} = 0.1$, and $\sigma_e^{(3)} = \sigma_i^{(3)} = 0.2$. A presynaptic neuron can form multiple synaptic connections with a single postsynaptic neuron.

Each neuron is an exponential integrate-and-fire model, in which the membrane potential follows the dynamics:

$$C_m \frac{dV_j^\beta}{dt} = -g_\beta \left( V_j^\beta - E_L \right) + g_\beta \Delta_T e^{\frac{V_j^\beta - V_T}{\Delta_T}} + I_j^\beta(t). \tag{23}$$

$g_\beta$ is the membrane conductance of neurons of type $\beta$. $C_m$ is the membrane capacitance. $E_L$ is the resting potential. $V_T$ is the spike initiation threshold, the voltage level at which the neuron begins to exhibit rapid depolarization, leading to spike generation. $\Delta_T$ is the sharpness parameter. It controls the steepness of the exponential rise in the membrane potential as the neuron approaches the threshold $V_T$. When $V_j^\beta$ exceeds a threshold $V_{th}$, the neuron emits a spike, and the membrane potential is then set to a fixed value $V_{re}$ for a duration of the refractory period $\tau_{ref}$. The total input current received by neuron $j$ is

$$\frac{I_j^\beta(t)}{C_m} = \sum_{\alpha \in \{f, i, e\}} \sum_{k=1}^{N_\alpha} \frac{J_{kj}^{\alpha\beta}}{\sqrt{N_e + N_i}} \sum_s \eta_\alpha \left( t - t_s^{\alpha k} \right) + \mu_\beta. \tag{24}$$

Here $J_{kj}^{\alpha\beta}$ is the synaptic weight from neuron $k$ to neuron $j$, where $\beta$ indicates the type of postsynaptic neuron $j$ and $\alpha$ indicates the type of presynaptic neurons. $\mu_\beta$ is the static current injected into neurons of type $\beta$. The times $t_s^{\alpha k}$ indicate the time of $s$th spike of neuron $k$ from the population $\alpha$. The postsynaptic current triggered by a single spike is

$$\eta_\alpha = \frac{1}{\tau_d^\alpha - \tau_r^\alpha} \begin{cases} e^{-\frac{t}{\tau_d^\alpha}} - e^{-\frac{t}{\tau_r^\alpha}} & t > 0, \\ 0 & t < 0, \end{cases} \tag{25}$$

where $\tau_r^\alpha$ and $\tau_d^\alpha$ are the synaptic rise and decay time constants, respectively, for population $\alpha$. The feedforward synapses from Layer 2 to Layer 3 consist of both fast and slow components $\eta_F(t) = 0.2 \cdot \eta_e(t) + 0.8 \cdot \eta_s(t)$, where $\eta_s(t)$ has the same form as Eq. (25) with a rise time constant $\tau_r^s = 2$ ms and a decay time constant $\tau_d^s = 100$ ms. All other parameters of neurons are provided in Table 1 and the connectivity parameters are provided in Table 2.

### Reporting summary
Further information on research design is available in the Nature Portfolio Reporting Summary linked to this article.

### Data availability
The synthetic data used in this study can be reproduced using the source code. Intracellular recordings of the membrane potential are presented in ref. 32 and available on Zenodo at https://zenodo.org/records/1304771. Neural recording data from V4 during the attention task are presented in ref. 54 and available on Fighshare at https://doi.org/10.6084/m9.figshare.16934326.v3. Neural recording data from PMd during the decision-making task are presented in ref. 56 and available on Figshare at https://doi.org/10.6084/m9.figshare.29052116.v1. Neural recording data from LIP during the decision-making tasks are presented in ref. 55 and are available on Figshare at https://doi.org/10.6084/m9.figshare.29604614.v1. Source data are provided with this paper.

### Code availability
The source code to reproduce the results of this study is available as a Python package on GitHub at https://github.com/engellab/DSRP.

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

## Acknowledgements

This work was supported by the Swartz Foundation (C.A.), the National Institutes of Health (NIH) grant R01EB026949 (T.A.E. and C.A.), NIH grant RF1DA055666 (T.A.E. and C.A.), NIH grant S10OD028632-01 (T.A.E. and C.A.), Alfred P. Sloan Foundation Research Fellowship (T.A.E.), NIH grant K99/R00NS092972 (C.C.), NIH grant NS121409 (C.C.), NIH grant NS122969 (C.C.), NIH grant NS135361 (C.C.), the Brain and Behavior Research Foundation (C.C.), and the Whitehall Foundation (C.C.). The authors thank A.K. Churchland and M.N. Shadlen for sharing the electrophysiological data for LIP, which are presented in ref. 55. The authors thank A. Banerjee for useful discussions and suggestions on the project. The authors thank M.N. Shadlen for thoughtful comments on the manuscript.

## Author contributions

C.A., C.C., and T.A.E. designed the research. C.A. and T.A.E. developed the theoretical framework and data analysis methods. C.A. performed analytical calculations, developed the code, performed computer simulations, and analyzed neural recording data. C.C. performed the experiments, spike sorting, and data curation for the PMd dataset. C.A., C.C., and T.A.E. wrote the paper.

## Competing interests

The authors declare no competing interests.
