## [Transparent Peer Review file · Nature Communications]

A doubly stochastic renewal framework for partitioning spiking variability

Corresponding Author: Professor Tatiana Engel

Version 0:

Reviewer comments:

Reviewer #1

(Remarks to the Author)

In this paper the authors propose a novel framework for analyzing neural spike trains, the doubly stochastic renewal (DSR) process. This is an extension of the popular doubly stochastic Poisson process, that can account for both variability of the firing rate and the (ir)regularity of neural firing that may vary widely among different neurons from very regular (like a pacemaker cell with a low CV) to burst-like firing (more irregular than a Poisson process) but also included the doubly stochastic Poisson process as a special case.

The authors introduce their framework and outline in particular how its parameters can be inferred from simulated or experimentally measures spike trains. In a first step, they derive simple formulas for the mean and variance of the spike count over a moderate to large time bin, assuming that the rate fluctuations are sufficiently slow. These formulas are expressed in terms of the first three central moments of the ISI density of the corresponding homogeneous renewal process. For the case of a Gamma distribution for the ISIs, the first three moments can be all expressed by the mean and the irregularity parameter ϕ (essentially, the squared coefficient of variation of the ISI), and the formulas for the count variance of the DSR process can be solely expressed by the size of the time bin, the variance of the fluctuating rate, and by the irregularity of the spike generator (quantified by ϕ). By considering count variances for tow different time bins, the authors are able to come up with a simple quadratic equation for the irregularity parameter that otherwise contains only measured statistics. They can in this way disentangle the contributions of spike generator and rate fluctuations - a notoriously difficult problem in computational neuroscience.

They contrast their model's performance with that of previously suggested models on simulated data (for which the ground truth is therefore known), illustrating the new framework's superiority, at least for the simulated examples. They then apply their framework to experimentally measured data from various brain regions, obtain in particular the parameter quantifying the irregularity of neural firing, and demonstrate that these results are in line with ideas about how the variability changes across cortical areas. They observe that spike irregularity remains maintained for individual neurons under different conditions. The authors finally simulate a recurrent network of spiking neurons and relate their irregularity parameter with changes in biophysical parameters and to the observations made on their experimental data.

There is a lot going on in this manuscript. The development of the framework of a doubly stochastic renewal model, the analytical calculation of mean and variance of the count for (not necessarily large) time windows, the formula for the extraction of the disorder parameter (without the necessity to know the rate fluctuations in advance) are already great and original contributions to the field. I dare to predict that this framework will be thankfully adopted and extended by many theory groups but, even more importantly, will be also used directly by experimental labs that explore the role of spike variability in neural signal processing.

The comparison to previous methods and the application to the experimental recordings of subthreshold membrane voltages are, according to my impression, solid consistency checks. More exciting again are the results for the irregularity parameter across different brain areas and stimulation conditions. I agree with many of the authors' conclusions but would like to see more discussion of the simple effect of the neural refractory period (see below).

In conclusion, I think this paper could make a substantial contribution to the field, certainly meriting publication in Nature Communication. I have a number of major and minor issues that should be addressed before the paper can be accepted for publication.

Major issues

1) Relation between high firing rates and low CV for cortical cells

I think there is a trivial component in the observed negative correlation: the neural refractory period. If the neural action potential has a finite width of several ms and there is an additional dead time in which the neuron cannot fire immediately a second spike, this limits the variability of ISIs that are on average only 10ms for a firing rate of 100Hz. This contribution to the observed effect should be thoroughly discussed.

2) Fairness in the comparison of the different models

It does not appear very convincing to me to test different procedures on synthetic data that are generated according to exactly one of the procedures (the DSR process). It is then not surprising that the one procedure (in this case the DSR process) performs best. The authors should put this result more modestly into perspective.

3) Renewal assumption for neural data and for stochastic integrate-and-fire models

On p.13 in the Discussion section the authors write

"The assumptions of our doubly stochastic renewal framework are consistent with phenomenological models of spiking neurons, such as the leaky integrate-and-fire model. Specifically, in both current-based and conductance-based models driven by the membrane potential fluctuations that are small compared to the distance from the resting potential to the firing threshold, the next spike time depends only on time since the last generated spike, which is equivalent to the renewal assumption."

Now, both for the leaky integrate-and-fire model as well as for experimental data it is known that ISIs can be correlated, which is in marked contrast to the claim made by the authors. In my opinion, it is not a good idea to oversell the overall applicability of the renewal process - it might be a useful approximation in many situations but there are a number of notable exceptions (experimental evidence is reviewed in Farkhooi et al. Phys. Rev. E 2009 and Avila-Akerberg & Chacron Exp Brain Res, 2011). Especially for integrate-and-fire models it has been shown that they easily generate ISI correlations, for instance, in the presence of adaptation currents (Liu & Wang J. Comput Neurosci 2001), low-pass filtered noise (Schwalger & Schimansky-Geier PRE 2008), narrowband noise (Bauermeister et al. PLoS Comp Biol. 2013), or by the combined effects of adaptation and correlated noise (Ramlow & Lindner PLoS Comp Biol. 2021).

I am not sure though whether ISI correlations might be even incorporated into the authors' 'renewal' model via the rate fluctuations in certain simple cases (say, for IF models driven by colored noise) - a point that should be definitely discussed by the authors. In this respect, it is probably also a good idea to emphasize that the suggested model differs strongly by the renewal theory put forward by Gerstner (e.g. Phys Rev E, 1995) that assumes strict dependence of the firing only on signals since the last spike time (but not before that time); for the suggested model, ISIs are not completely independent when the rate fluctuations are correlated!

4) Correlation between subthreshold membrane potential and firing rate fluctuations.

A better motivation for the positive correlation between subthreshold membrane voltage and firing rate should be given. Apparently, for the occurrence of a spike, the subthreshold voltage has to cross the threshold, hence, it has to be high at least for a certain fraction of time when there is a spike in the time bin T . Thus, a positive correlation between spike rate and membrane voltage can be expected on somewhat trivial grounds. I got the feeling that the authors try to eliminate this trivial correlation by excluding an interval around each action potential from the statistical analysis. What else is then the reason for the expected correlation? Can this be expected also for a model, e.g. the leaky integrate-and-fire model, later used in the paper? Some more motivation and explanation of this issue is required.

5) Supplemental information, Proof of theorem 2

I do not think this is a mathematically rigorous proof. First of all, it is not clear at which time the function $\lambda(t)$ has to be taken in the final relation

$$E[N_T | \lambda] = T/(1/\lambda) = \lambda T$$

Secondly, for the special case of an inhomogeneous Poisson process, λT has obviously to be replaced by $\int_{t-T}^t \lambda(s) ds$ for any truly time-dependent rate $\lambda(t)$. Hence, for any truly changing $\lambda(t)$, it can only be approximately constant over the time bin T and thus, the resulting relationship can only be approximately true. In conclusion, either one should write

$$E[NT | \lambda] \approx \lambda T$$

and clearly state what λ is (An average over the considered bin? The value at the beginning, the middle or the end of the bin?). Or, one should write

$$E[NT | \lambda] = \lambda T + \mathcal{O}(\dots)$$

with a similar statement on what λ should be taken and the proof about the order term.

Here, I am not sure about the exact nature of the order term. It might be expressed by the temporal derivative of the rate (possibly averaged over the bin?), to reflect a correction to an infinitely slow rate fluctuation (a constant). The correction term should make clear what the exact condition is under which the approximate asserted relation holds true.

Minor issues

p.17 Synthetic data generation

The authors should state the initial value of λ to complete the model of the rate fluctuations.

p.17 The authors write

"To ensure that Eq. 5 holds, we require $T > 1/E[\lambda]$. Since we assume the firing rate is constant within a bin, we also need to choose a bin size as small as possible. To satisfy both conditions, we set $T = 1/E[\lambda]$ for each neuron ..."

It is not possible to have $T > 1/E[\lambda]$ and $T = 1/E[\lambda]$ at the same time.
Reformulate!

p.18 The authors write:

"we discretize the membrane potential range between -68 mV and -40 mV into 1 mV bins and average the voltage and spike counts data within these bins."

Not clear to me, how you average spike count data in voltage bins. Explain and/or reformulate the sentence.

Suppl. information

p.3 I find it unfortunate to denote the Laplace transform with an asterisk (in the superscript) that also indicates the convolution operation (between functions). I would suggest to use a hat or a tilde to indicate the Laplace transform.

p.3 The authors write

"One can show that we have [1, 2]"

This relation has been known for ages and does not have 'to be shown'.

Typos

p.9, l. 245: changing firing rate

λ is sometimes written in bold face, sometimes not - not clear, why.
I suspect that $\lambda(t)$ without **λ** is meant to be a sample of the stochastic process.
If this is the case, it should be spelled out somewhere (best, in the beginning!)

p.16, l. 493 "On the other hand, if α is very close to 1 ..."

Where is the "On the one hand, ..." ?

Suppl. information

p.3 "such that for $\text{Re}(s) >$ "

Do not use italic font for the real part.

(Remarks on code availability)

Reviewer #2

(Remarks to the Author)

Aghamohammadi C et al., A doubly stochastic renewal framework for partitioning spiking variability

The authors put forward a model to account for spike count variance in neural responses by partitioning the variance into variance in the underlying rate and variance in the ISI distribution. They show that their model recovers estimates of spiking variability accurately, under the conditions examined in the manuscript. Furthermore, the spiking variability estimate is approximately constant for a neuron under different conditions. Finally, they show that spiking variability in a network model is related to the network connectivity.

In general, I quite like this paper. It has important methodological implications for the further development of models, including non-linear dimensionality reduction models, on which this group works. The approach may also be useful for exploring changes in variability across the brain, which was done to a limited extent in the manuscript, but could be pursued further in the future. I have several general comments, which mostly relate to further exploring the regime in which this model is appropriate.

Comments

1. I'm not sure I know the value that was used for T ? What interval was used to estimate mean and variance of spike counts? The interval over which λ is constant likely varies considerably from area to area, and even within neurons across time. For example, neurons often have a phasic initial response, followed by a more prolonged response. Understanding the value of T used will be important to understanding where this method can be usefully applied. In the hippocampus, for example, this approach is not likely to be valid. But it may also vary in validity in other areas.

2. "In all areas, neurons with high firing rates had low spiking irregularity: among 25% neurons with the highest firing rates, virtually..." The refractory period can substantially decrease the variability in spike counts, by negatively correlating spike probabilities. Some consideration and discussion of the effect of absolute and relative refractory periods would be useful to tie this approach to the biophysics of single neurons.

3. I think it would be more useful to have Figure 3 C as a scatter plot. At least for ϕ and spiking irregularity.

4. It would be useful to discuss this approach, and the results shown across different brain areas, in the context of these older papers (both of which were referenced):

Maimon, G. & Assad, J. A. Beyond Poisson: increased spike-time regularity across primate parietal cortex. *Neuron* 62, 426–440 (2009).

Shinomoto, S. et al. Relating neuronal firing patterns to functional differentiation of cerebral cortex. *Plos Comput Biol* 5 (2009).

Also, it would be useful to add some discussion of how these measures of variability relate to the time-scale estimates that other groups have been looking at:

<https://pubmed.ncbi.nlm.nih.gov/33431695/>

<https://pubmed.ncbi.nlm.nih.gov/32839338/>

(Remarks on code availability)

Reviewer #3

(Remarks to the Author)

This manuscript presents a new methodology to partition the total spiking variability of neurons into firing rate fluctuations and spiking irregularity. Previous works have addressed this issue with alternative methods, and the manuscript shows that the new method is superior because it can correctly parse firing rate fluctuations across trials and across time within trial. The manuscript presents the mathematical derivations supporting their method, which includes a smart transformation to an "operational" dimension, and the mathematical comparison with existing methods. The method is validated against surrogate data from diffusion models where the contributions of the various forms of variability is known. It is also validated using intracellularly recorded neurons, in a clever approach that uses voltage traces as a proxy to derive the latent firing rate variability. Then, the method is applied to three datasets of extracellularly recorded neurons in different cortical areas during various laboratory tasks. Several results in the literature are replicated, such as changes in neural variability across the cortical hierarchy, or depending on behavioral state. Finally, a computational network model is used to show that results compatible with the observations in the datasets can be obtained from spiking neural network simulations.

In my opinion, this manuscript presents a relevant new methodology to analyze neural recordings in behaving animals and provide a more rigorous way to separate the sources of neural variability in order to facilitate the interpretation of data. The mathematical derivations are elegant and clearly presented, and their strength is nicely demonstrated with a variety of validation methods: from analytical comparison with competitor methods, to direct validation and comparison against ground truth surrogate data, and to an elegant validation with intracellularly recorded voltage traces. I found more shallow the application of the method to existing datasets and network simulations. My major concerns are:

1) There is no clear example where the application of this new method proves crucial to gain new insight into brain function. In particular, previous methods already concluded that neural variability declines in higher cortical areas (Maimon et al *Neuron* 2009). On the other hand, the invariance of neural variability with attentional conditions could be interesting but is not fully discussed. The increase of neural variability in the decision period relative to the fixation period observed is presented in the Discussion as contrary to existing literature (citing perceptual, motor and attention tasks), but not in relation to more similar decision and memory tasks, which go in the same direction (Churchland et al. *Neuron* 2011; Compte et al. *J Neurophysiol* 2003). The overall message from this section is that the method is able to replicate previous results, but does not offer a specific advantage to correct previous misinterpretations.

2) It is unclear how the model simulations help to support the new methodology, especially since the simulations do not incorporate variability in the firing rate (see e.g. Ostojic *Nat Neurosci* 2014, 10.1038/nn.3658) so there is not a problem of variability attribution. Also, the fact that balanced excitatory-inhibitory networks with probabilistic connectivity structures have a broad distribution of firing rates and neural variability, comparable to in situ data, is known (van Vreeswijk and Sompolinsky, *Science* 1996; Neural Comput 1998; Roxin et al. *J Neurosci* 2011). On the other hand, the manipulations to alter neural irregularity in the network are over-simplified: considering that a focus is on changes in conductance it would appear necessary to consider conductance-based synapses to understand the modulations expected in more realistically connected networks (lines 393-394).

3) Instead, the modeling frustrates the expectations of the reader, since at several points in the manuscript it is speculated that changes in spiking irregularity in the cortical hierarchy could relate to functional specialization, but this is never tested in simulations. Instead of spontaneous activity without any specific task structure in the simulation, known tasks could be simulated to address how changes in network dynamics related to specific tasks could modulate irregularity. The model could generate function by putting it into and out of some attractor dynamics (e.g. Hansel and Mato, *J Neurosci* 2013, 10.1523/JNEUROSCI.3455-12.2013) or applying some attentional modulation (e.g. Huang et al. *Neuron* 2019 10.1016/j.neuron.2018.11.034).

Minor comments:

1) line 209 citation to ref. 80 seems to forget the classic studies that first proposed this calculation: Ricciardi 1977, Amit and Tsodyks 1981, Amit and Brunel 1997

2) line 692: please define x_i , it is necessary to understand the following equation. Is it $x_i = i/N_\alpha$?

3) line 693: this probability function is only approximately properly normalized if $\sigma \ll 1$. Also, could the summation be restricted to be from $k=-1$ to $k=1$, when $\sigma \ll 1$ and $-1 < r < 1$?

4) line 702: $J_{kj}^\alpha \beta$ has never been defined. Same for μ_β (which is confusing because μ has been used for something else before), g_β , Δ_T , V_T , E_L

(Remarks on code availability)

The source code to reproduce the results of this study will be made available on GitHub upon publication.

Reviewer #4

(Remarks to the Author)

Spike rates have long been considered the fundamental currency of neural computations. The noisiness of neuronal spiking is traditionally modeled as a Poisson process. But the observed spiking patterns in many areas of the brain, especially in association and premotor areas, significantly deviate from the predictions of a Poisson process. And yet, there is no established model to accurately partition variance arising from underlying rates vs. the variance that comes from the spike generation process itself. The current manuscript provides an impressive mathematical framework for accomplishing that. This study is a tour de force:

- The mathematical approach is elegant and well thought out,
- The model is thoroughly validated at multiple levels — from synthetic data; from intracellular recordings; from single unit data from sensory, association and motor cortices; and from in silico modeling
- The authors present novel insights from this modeling approach — establishing how spike generation process varies between sensory, association and motor cortices; and how it can even vary for the same neuron across task epochs
- The narrative is very well written and easy to follow
- A clear recipe is provided for the field to implement this model with their own data.
- The approach presented has the potential to be widely used in systems neuroscience

I have no major concerns about the manuscript. Below, I list a few minor suggestions that the authors could consider for improving the manuscript:

Figure 3c: The description is hard to follow. It would be helpful if this panel is introduced by a topic sentence. Also, a horizontal line at 1 could help readers make a quick visual assessment of how close ϕ is to 1 for each neuron.

Ln 227: The claim that the spiking becomes "... progressively more ..." regular is debatable. There appears to be a bimodality to the LIP ϕ distribution, which places subpopulations of LIP neurons on either side of PMd ϕ distribution.

Ln 242: Similarly, the claim that virtually no neurons reside in the top right corner of Figure 4b can be softened.

Ln 63/64: It is hard to distinguish $\mathbf{\lambda}$ from λ , especially in print

(Remarks on code availability)

Version 1:

Reviewer comments:

Reviewer #1

(Remarks to the Author)

The authors have addressed all my concerns and I now fully support the publication of this important paper in Nature Communications.

(Remarks on code availability)

Reviewer #2

(Remarks to the Author)

The authors have addressed my concerns. I have no further comments.

(Remarks on code availability)

Reviewer #3

(Remarks to the Author)

I am fully satisfied with the revisions of this manuscript. I congratulate the authors for this excellent paper.

(Remarks on code availability)

Reviewer #4

(Remarks to the Author)

The authors have comprehensively addressed all reviewer comments. My assessment of the rigor and potential impact of the original manuscript itself was already highly positive and the revisions further strengthen both. I have no remaining concerns.

(Remarks on code availability)

Point-by-point responses.

Reviewer #1:

In this paper the authors propose a novel framework for analyzing neural spike trains, the doubly stochastic renewal (DSR) process. This is an extension of the popular doubly stochastic Poisson process, that can account for both variability of the firing rate and the (ir)regularity of neural firing that may vary widely among different neurons from very regular (like a pacemaker cell with a low CV) to burst-like firing (more irregular than a Poisson process) but also included the doubly stochastic Poisson process as a special case.

The authors introduce their framework and outline in particular how its parameters can be inferred from simulated or experimentally measured spike trains. In a first step, they derive simple formulas for the mean and variance of the spike count over a moderate to large time bin, assuming that the rate fluctuations are sufficiently slow. These formulas are expressed in terms of the first three central moments of the ISI density of the corresponding homogeneous renewal process. For the case of a Gamma distribution for the ISIs, the first three moments can be all expressed by the mean and the irregularity parameter ϕ (essentially, the squared coefficient of variation of the ISI), and the formulas for the count variance of the DSR process can be solely expressed by the size of the time bin, the variance of the fluctuating rate, and by the irregularity of the spike generator (quantified by ϕ). By considering count variances for two different time bins, the authors are able to come up with a simple quadratic equation for the irregularity parameter that otherwise contains only measured statistics. They can in this way disentangle the contributions of spike generator and rate fluctuations - a notoriously difficult problem in computational neuroscience.

They contrast their model's performance with that of previously suggested models on simulated data (for which the ground truth is therefore known), illustrating the new framework's superiority, at least for the simulated examples. They then apply their framework to experimentally measured data from various brain regions, obtain in particular the parameter quantifying the irregularity of neural firing, and demonstrate that these results are in line with ideas about how the variability changes across cortical areas. They observe that spike irregularity remains maintained for individual neurons under different conditions. The authors finally simulate a recurrent network of spiking neurons and relate their irregularity parameter with changes in biophysical parameters and to the observations made on their experimental data.

There is a lot going on in this manuscript. The development of the framework of a doubly stochastic renewal model, the analytical calculation of mean and variance of the count for (not necessarily large) time windows, the formula for the extraction of the disorder parameter (without the necessity to know the rate fluctuations in advance) are already great and original contributions to the field. I dare to predict that this framework will be thankfully adopted and extended by many theory groups but, even more importantly, will be also used directly by experimental labs that explore the role of spike variability in neural signal processing. The comparison to previous methods and the application to the experimental recordings of subthreshold membrane voltages are, according to my impression, solid consistency checks. More exciting again are the results for the irregularity parameter across different brain areas and stimulation conditions. I agree with many of the authors' conclusions but would like to see more discussion of the simple effect of the neural refractory period (see below).

In conclusion, I think this paper could make a substantial contribution to the field, certainly meriting publication in Nature Communication. I have a number of major and minor issues that should be addressed before the paper can be accepted for publication.

Reply:

We thank the reviewer for this enthusiastic assessment and recognition of the broad significance of our work.

Major issues

1) Relation between high firing rates and low CV for cortical cells

I think there is a trivial component in the observed negative correlation: the neural refractory period. If the neural action potential has a finite width of several ms and there is an additional dead time in which the neuron cannot fire immediately a second spike, this limits the variability of ISIs that are on average only 10ms for a firing rate of 100Hz. This contribution to the observed effect should be thoroughly discussed.

Reply:

The refractory period can indeed contribute to the negative correlation between the mean firing rate and spiking irregularity. Prompted by the reviewer's comment, we performed two additional analyses to verify that the refractory period is not the sole source of the negative correlation observed in the data (Fig. 4). First, we estimated the mean firing rate above which the refractory period substantially impacts the spiking irregularity. Second, we systematically removed high firing-rate neurons from experimental data to verify that the negative correlation between the firing rate and spiking irregularity is not solely driven by spiking regularity of high firing-rate neurons.

First, we estimate the firing rate above which the refractory period significantly impacts spiking irregularity. We consider an integrate-and-fire type neuron that fires a spike when the voltage reaches a threshold V_{th} , upon which the voltage is reset to V_r after a fixed refractory period RP . The neuron receives a presynaptic input spike train following a homogeneous renewal process with a constant firing rate λ_{in} . Each presynaptic spike increments the membrane potential by an amount J . In the stationary regime, we can decompose the postsynaptic interspike interval (ISI) into the refractory period and a random component: $ISI = RP + X$. The probability distribution of random variable X generally depends on parameters such as neuron properties, input statistics, and also RP . Let us assume that $(V_{th} - V_r)/J \gg 1$, meaning that the neuron needs to receive $n \gg 1$ presynaptic spikes to generate the next spike after

a reset. We can then write $X = t_0 + \sum_{i=1}^{n-1} t_i$, where t_0 is time between the end of RP and arrival of the first presynaptic spike, and t_i is the ISI between the i -th and $(i+1)$ -th presynaptic spikes. Since the input process is renewal, only t_0 in X is influenced by the refractory period, and X is approximately independent of RP for large n . For constant input firing rate, we can compute the spiking irregularity using ISI moments in real time, instead of operational time, to obtain $\phi = \frac{\sigma_{ISI}^2}{\mu_{ISI}^2} = \frac{\sigma_X^2}{(RP + \mu_X)^2} = \frac{CV_X^2}{(\frac{RP}{\mu_X} + 1)^2}$. Here CV_X is the coefficient of variation of X , and the average firing rate of the neuron is $fr = \frac{1}{RP + \mu_X}$. This equation shows that spiking irregularity decreases with increasing RP . In the limit $\mu_X \gg RP$ (equivalently, $RP \ll \frac{1}{fr}$), the refractory period has negligible effect and $\phi \simeq CV_X^2$. In the opposite limit $\mu_X \ll RP$ (equivalently, $RP \simeq \frac{1}{fr}$), the refractory period dominates and $\phi \simeq 0$. In between these two extremes, both RP and X jointly determine the spiking irregularity. The refractory period (including the duration of the action potential) of neurons typically ranges from approximately 4 to 12 ms [1]. Hence, the firing rate above which the refractory period dominates spiking irregularity ranges from $\frac{1}{12\text{ms}} \approx 83\text{ Hz}$ to $\frac{1}{4\text{ms}} \approx 250\text{ Hz}$.

Second, we examined the potential effect of the refractory period on the negative correlation between the firing rate and spiking irregularity in experimental data. The correlation remained significant when restricting analysis to neurons with firing rates below 83 Hz (LIP $r = -0.51$, $p=0.0001$; V4 $r = -0.24$, $p=0.0001$) and even below 50 Hz, typical of experimental data (LIP $r = -0.48$, $p=0.0015$; V4 $r = -0.22$, $p=0.0002$). Therefore, the negative correlation between firing rate and spiking irregularity does not arise solely from the refractory period effects of high firing-rate neurons.

We describe these analyses on lines 260–261 in revised Results and in new Supplementary Note 1.5.

[1] Sardi, Shira, et al. "Long anisotropic absolute refractory periods with rapid rise times to reliable responsiveness." *Physical Review E* 105.1 (2022): 014401.

2) Fairness in the comparison of the different models

It does not appear very convincing to me to test different procedures on synthetic data that are generated according to exactly one of the procedures (the DSR process). It is then not surprising that the one procedure (in this case the DSR process) performs best. The authors should put this result more modestly into perspective.

Reply:

The reviewer raises a valid point that any method is expected to perform best on data generated according to its assumptions, as is the case for the DSR method in our synthetic data tests (Fig. 2). Nevertheless, the comparison of our DSR method with Deterministic Time Rescaling (DTR) and Minimum Ratio (MR) methods is fair. DTR assumes an inhomogeneous renewal process as a generative model, which is a special case of our DSR model with zero trial-to-trial firing rate variability. Thus, data generated by the DSR model with zero trial-to-trial firing rate variability matches the assumptions of the DTR method. Accordingly, the accuracy of the DTR method is high when the trial-to-trial firing rate variability is low, in the regime consistent with its assumptions (Fig. 2b,c). In contrast, the MR method is based on heuristics lacking any generative point process model. Therefore, it is not possible to evaluate the accuracy of this method in a setting that aligns with its assumptions. In the absence of a generative model, it is not even clear what quantity the MR method estimates, consistent with its erratic performance in estimating parameter φ in the DSR model (Fig. 2b,c).

We clarify this point on lines 183–189 in the revised Results.

3) Renewal assumption for neural data and for stochastic integrate-and-fire models

*On p.13 in the Discussion section the authors write "The assumptions of our doubly stochastic renewal framework are consistent with phenomenological models of spiking neurons, such as the leaky integrate-and-fire model. Specifically, in both current-based and conductance-based models driven by the membrane potential fluctuations that are small compared to the distance from the resting potential to the firing threshold, the next spike time depends only on time since the last generated spike, which is equivalent to the renewal assumption." Now, both for the leaky integrate-and-fire model as well as for experimental data it is known that ISIs can be correlated, which is in marked contrast to the claim made by the authors. In my opinion, it is not a good idea to oversell the overall applicability of the renewal process - it might be a useful approximation in many situations but there are a number of notable exceptions (experimental evidence is reviewed in Farkhooi et al. *Phys. Rev. E* 2009 and Avila-Akerberg & Chacron *Exp Brain Res*, 2011). Especially for integrate-and-fire models it has been shown that they easily generate ISI correlations, for instance, in the presence of adaptation currents (Liu & Wang *J. Comput Neurosci* 2001), low-pass filtered noise (Schwalger & Schimansky-Geier *PRE* 2008), narrowband noise (Bauermeister et al. *PLoS Comp Biol.* 2013), or by the combined effects of adaptation and correlated noise (Ramlow & Lindner *PLoS Comp Biol.* 2021). I am not sure though whether ISI correlations might be even incorporated into the authors' 'renewal' model via the rate fluctuations in certain simple cases (say, for IF models driven by colored noise) - a point that should be definitely discussed by the authors.*

Reply:

We agree that the quoted text from our original manuscript was misleading and thank the reviewer for bringing up the highly relevant literature on ISI correlations. Indeed, serial correlations between ISIs are observed in data (Farkhooi et al. *Phys. Rev. E* 2009; Engel et al. *J. Neurophysiol.* 2008) and can arise from correlated input or firing-rate adaptation in mechanistic integrate-and-fire models (Liu & Wang, *J Comput Neurosci* 2001; Schwalger & Schimansky-Geier, *PRE* 2008; Bauermeister et al., *PLoS Comp Biol.* 2013; Ramlow & Lindner, *PLoS Comp Biol.* 2021).

While our DSR model assumes that ISIs are independent in *operational time*, it generates serial ISI correlations in *real time* through temporally correlated instantaneous firing rate. An important question is in what regime can the DSR framework capture serial ISI correlations arising from correlated input or adaptation in mechanistic integrate-and-fire models? ISIs are approximately uncorrelated in real time in the low firing-rate regime, where the

timescale of the correlated input or adaptation is much shorter than a typical ISI. Serial ISI correlations arise in the high firing-rate regime, where the timescale of the correlated input or adaptation is longer than a typical ISI. Thus, we tested whether in this regime, our DSR model can capture serial ISI correlations through the slow dynamics of the instantaneous firing rate with approximately independent ISIs in operational time.

We used a leaky integrate-and-fire (LIF) neuron model driven by exponentially correlated Gaussian noise (Schwalger & Schimansky-Geier, *PRE* 2008). The dynamics of the membrane potential $V(t)$ is described by the equation $\tau_m \frac{d}{dt} V = -V + RI_s(t)$. The neuron fires a spike when the membrane potential reaches a threshold V_{th} , after which the membrane potential is reset to V_r . The synaptic input current $I_s(t)$ is an Ornstein-Uhlenbeck process: $\tau_s \frac{d}{dt} I_s = -I_s + \mu + \sigma \xi(t)$, with white Gaussian noise $\xi(t)$ that obeys $\langle \xi(t)\xi(t') \rangle = \delta(t - t')$. We consider input with a long correlation time by setting $\tau = \frac{\tau_s}{\tau_m} = 200$ and $D = \frac{\sigma^2 R^2}{2\tau_m(V_{th} - V_r)^2} = 0.3$. With these parameters, the neuron generates spikes showing strong serial ISI correlations (Pearson correlation coefficient between adjacent ISIs $r=0.97$).

We tested whether our DSR framework can capture these serial ISI correlations using the same analysis that we developed for intracellular voltage recordings (Fig. 3). First, we estimate an empirical function relating the average subthreshold voltage to firing rate. Then, we use this function to compute the instantaneous firing rate from the subthreshold voltage. Finally, we use this estimated instantaneous firing rate to map spikes from real to operational time via time rescaling. The ISI correlations in the operational time were considerably reduced (Pearson correlation coefficient between adjacent ISIs $r=0.11$). This example illustrates that our DSR model can incorporate ISI correlations in real time via temporally correlated rate fluctuations, when the timescale of the correlated input or adaptation is shorter than a typical ISI.

We removed the text quoted by the review and added a discussion of ISI correlations on lines 455–461 in revised Discussion and in new Supplementary Note 1.8.

4) Correlation between subthreshold membrane potential and firing rate fluctuations.

A better motivation for the positive correlation between subthreshold membrane voltage and firing rate should be given. Apparently, for the occurrence of a spike, the subthreshold voltage has to cross the threshold, hence, it has to be high at least for a certain fraction of time when there is a spike in the time bin T. Thus, a positive correlation between spike rate and membrane voltage can be expected on somewhat trivial grounds. I got the feeling that the authors try to eliminate this trivial correlation by excluding an interval around each action potential from the statistical analysis. What else is then the reason for the expected correlation? Can this be expected also for a model, e.g. the leaky integrate-and-fire model, later used in the paper? Some more motivation and explanation of this issue is required.

Reply:

This analysis is grounded in previous theoretical studies showing that, for a variety of LIF neuron models, the average firing rate and membrane potential are related by a power law function [1,2]. This relation has been also verified experimentally in many cases [3,4]. We base our analysis on these previous findings, which we replicate both in experimental data (Fig. 3b) and LIF model (new Supplementary Note 1.8).

We clarified this point on lines 199–204 in revised Results and in new Supplementary Note 1.8.

[1] Hansel, David, and Carl van Vreeswijk. "How noise contributes to contrast invariance of orientation tuning in cat visual cortex." *Journal of Neuroscience* 22.12 (2002): 5118-5128.

[2] Miller, Kenneth D., and Todd W. Troyer. "Neural noise can explain expansive, power-law nonlinearities in neural response functions." *Journal of Neurophysiology* 87.2 (2002): 653-659.

[3] Priebe, Nicholas J., and David Ferster. "Inhibition, spike threshold, and stimulus selectivity in primary visual cortex." *Neuron* 57.4 (2008): 482-497.

[4] Priebe, Nicholas J., et al. "The contribution of spike threshold to the dichotomy of cortical simple and complex cells." *Nature Neuroscience* 7.10 (2004): 1113-1122.

5) Supplemental information, Proof of theorem 2

I do not think this is a mathematically rigorous proof. First of all, it is not clear at which time the function $\lambda(t)$ has to be taken in the final relation

$$E[N_T | \lambda] = T/(1/\lambda) = \lambda T$$

Secondly, for the special case of an inhomogeneous Poisson process, λT has obviously to be replaced by $\int_{t_0}^{t_0+T} \lambda(s) ds$ for any truly time-dependent rate $\lambda(t)$. Hence, for any truly changing $\lambda(t)$, it can only be approximately constant over the time bin T and thus, the resulting relationship can only be approximately true. In conclusion, either one should write

$$E[N_T | \lambda] \approx \lambda T$$

and clearly state what λ is (An average over the considered bin? The value at the beginning, the middle or the end of the bin?). Or, one should write

$$E[N_T | \lambda] = \lambda T + \mathcal{O}(\dots)$$

with a similar statement on what λ should be taken and the proof about the order term.

Here, I am not sure about the exact nature of the order term. It might be expressed by the temporal derivative of the rate (possibly averaged over the bin?), to reflect a correction to an infinitely slow rate fluctuation (a constant). The correction term should make clear what the exact condition is under which the approximate asserted relation holds true.

Reply:

We agree with this comment and have revised the theorems to ensure rigor. We updated Theorem 2 restricting its statement to stochastic λ that is constant within a time bin. We added new Lemma 1 addressing the case where the firing rate is a deterministic function of time. For an equilibrium inhomogeneous renewal point process $\{g(\cdot), \lambda(t)\}$, where the firing rate $\lambda(t)$ is a deterministic function of time that does not vary across realizations, we show that the expected number of spikes $N_T(t_0)$ in the time bin $[t_0, t_0 + T]$ is $E[N_T(t_0)] = \int_{t_0}^{t_0+T} \lambda(t) dt$. We then introduce

Theorem 3 extending the results of Lemma 1 to cases where the firing rate is stochastic across realizations. For an equilibrium doubly stochastic renewal point process $\{g(\cdot), \lambda(t)\}$, where the firing rate $\lambda(t)$ is stochastic function of time varying from trial-to-trial, we show that $E[N_T(t_0)|\lambda(t)] = \int_{t_0}^{t_0+T} \lambda(t) dt$. Using Theorem 3, we then derive

Corollary 1, which applies Theorem 3 to the case where the stochastic parameter λ is constant within each time bin, thus rederiving the result of Theorem 2. Finally, we revised Theorem 4 restricting its statement to the case of stochastic λ that is constant within a time bin.

We revised Theorem 2 and Theorem 4 and added a new lemma, theorem, and corollary in the Supplementary Note 1.1.

Minor issues

p.17 Synthetic data generation

The authors should state the initial value of λ to complete the model of the rate fluctuations.

Reply:

The initial firing rate values were set to 10 Hz in Fig. 1 and 30 Hz in Fig. 2. We have clarified this detail on lines 584–585 in revised Methods.

p.17 The authors write

"To ensure that Eq. 5 holds, we require $T > 1/E[\lambda]$. Since we assume the firing rate is constant within a bin, we also need to choose a bin size as small as possible. To satisfy both conditions, we set $T = 1/E[\lambda]$ for each neuron ..."

It is not possible to have $T > 1/E[\lambda]$ and $T = 1/E[\lambda]$ at the same time.

Reformulate!

Reply:

Our original manuscript had a typo. In all analyses, we set $T = 2/E[\lambda]$ to satisfy both conditions. **We fixed this typo on line 591 in revised Methods.**

p.18 The authors write:

"we discretize the membrane potential range between -68 mV and -40 mV into 1 mV bins and average the voltage and spike counts data within these bins."

Not clear to me, how you average spike count data in voltage bins. Explain and/or reformulate the sentence.

Reply:

We segment the recorded voltage trace in $\Delta t = 50$ ms time bins. For each time bin i , we compute the average membrane potential V_i and the number of spikes N_i within that time bin. We then plot $N_i/\Delta t$ versus V_i for all time bins (blue dots in Fig. 3b). Next, we divide the voltage range $[-68, -40]$ mV into $\Delta V = 1$ mV voltage bins ΔV_k ($k = 1, 2, \dots, 28$). We find the set of all data points falling within k th bin: $S_k = [\forall i: V_i \in \Delta V_k]$ and compute the average firing rate $r_k = \frac{1}{|S_k| \Delta t} \sum_{i \in S_k} N_i$ and the average voltage $V_k = (-68 + k\Delta V)/2$ in that bin. Finally, we fit

this average relationship with a spline to approximate the dependence of the instantaneous firing rate r_k on subthreshold membrane potential V_k with the deterministic function $f(v)$.

We clarified this procedure on lines 604–616 in revised Methods.

Suppl. information

p.3 I find it unfortunate to denote the Laplace transform with an asterisk (in the superscript) that also indicates the convolution operation (between functions). I would suggest to use a hat or a tilde to indicate the Laplace transform.

Reply:

We changed the notation to indicate the Laplace transform with a hat.

p.3 The authors write

"One can show that we have [1, 2]"

This relation has been known for ages and does not have 'to be shown'.

Reply:

We changed this text to read "It holds ..." in Supplementary Notes.

Typos

p.9, l. 245: changing faring rate

Reply:

We fixed the typo.

λ is sometimes written in bold face, sometimes not - not clear, why.

I suspect that $\lambda(t)$ without $\mathbf{\lambda}$ is meant to be a sample of the stochastic process.

If this is the case, it should be spelled out somewhere (best, in the beginning!)

Reply:

We spell out this difference in notation for the stochastic process versus a sample from this process **on lines 63–64 in revised Results.**

p.16, l. 493 "On the other hand, if α is very close to 1 ..."

Where is the "On the one hand, ..." ?

Reply:

We replaced “On the other hand” with “However”.

Suppl. information

p.3 "such that for $Re(s) >$ "

Do not use italic font for the real part.

Reply:

We fixed the notation.

Reviewer #2:

Aghamohammadi C et al., A doubly stochastic renewal framework for partitioning spiking variability

The authors put forward a model to account for spike count variance in neural responses by partitioning the variance into variance in the underlying rate and variance in the ISI distribution. They show that their model recovers estimates of spiking variability accurately, under the conditions examined in the manuscript. Furthermore, the spiking variability estimate is approximately constant for a neuron under different conditions. Finally, they show that spiking variability in a network model is related to the network connectivity.

In general, I quite like this paper. It has important methodological implications for the further development of models, including non-linear dimensionality reduction models, on which this group works. The approach may also be useful for exploring changes in variability across the brain, which was done to a limited extent in the manuscript, but could be pursued further in the future. I have several general comments, which mostly relate to further exploring the regime in which this model is appropriate.

Reply:

We thank the reviewer for the positive assessment and recognition of the broad significance of our work, as well as for thoughtful suggestions that helped us improve the paper.

1. I'm not sure I know the value that was used for T ? What interval was used to estimate mean and variance of spike counts? The interval over which λ is constant likely varies considerably from area to area, and even within neurons across time. For example, neurons often have a phasic initial response, followed by a more prolonged response. Understanding the value of T used will be important to understanding where this method can be usefully applied. In the hippocampus, for example, this approach is not likely to be valid. But it may also vary in validity in other areas.

Reply:

To ensure that Eq. 5 holds, we require $T > 1/E[\lambda]$. Since we assume the firing rate is constant within a bin, we also need to choose a bin size as small as possible. To satisfy both conditions, we set $T = 2/E[\lambda]$ for each neuron in experimental and synthetic data, where $E[\lambda]$ is the average firing rate of the neuron over the analysis period. Thus, the bin size depends on the neuron's average firing rate and varies across neurons.

Prompted by the reviewer's comment, we performed two additional analyses to verify that our inference method is 1) robust and largely insensitive to bin size across a wide range of values, and 2) remains reliable even in the presence of rapid changes in firing rate.

First, we tested the robustness of our inference method to the specific choice of bin size. We used synthetic data generated from doubly stochastic renewal point processes with known ground truth φ . Specifically, we chose $g(\cdot)$ to be a gamma distribution, and the firing rate $\lambda(t)$ on each trial sampled from a drift-diffusion process (as in Fig. 2 lower row, trial duration = 2 s, number of trials = 100, upper threshold = 40 Hz, lower threshold = 10 Hz,

$D = 10 \frac{\text{Hz}^2}{\text{ms}}$). We estimate φ with our method from synthetic data, with 30 independent runs for each value of ground truth φ . We denote the bin size used in the paper as $T_0 = 2/E[\lambda]$, and vary the bin size within the range from

$(1/2) \cdot T_0$ to $(11/8) \cdot T_0$. We find that our estimation method remains highly accurate for this entire range of bin sizes (Supplementary Fig. 1).

Second, we tested the robustness of our method to rapid firing rate changes at certain times that are much faster than the chosen bin size $T = 2/E[\lambda]$. We use synthetic data generated from the doubly stochastic renewal point processes with known ground truth φ and the firing rate $\lambda(t)$ on each trial sampled from the stepping model: the firing rate starts at 10 Hz and jumps to 40 Hz at a random time on each trial. We have 50 trials that last 2 seconds. We estimate φ with our method from synthetic data, with 30 independent runs for each value of ground truth φ . We vary the bin size within the range from $(1/2) \cdot T_0$ to $(11/8) \cdot T_0$. We find that our estimation method remains accurate in this scenario where the firing rate exhibits occasional rapid changes in each trial (Supplementary Figs. 2,3).

Intuitively, the estimation method remains robust if rapid changes in the firing rate are relatively rare. Suppose each trial lasts 2 s and the average firing rate is approximately 20 Hz. We then choose a bin size of 100 ms, resulting in 20 bins per trial. If the firing rate jumps only once per trial, about 5% of bins include a jump in firing rate. Thus, across 20 trials, rapid firing-rate changes occur in only one or two trials on average, with negligible impact on the estimation accuracy. The specific case where neuronal firing rates show a phasic initial response, followed by a more prolonged response, only involves a small number of jumps in the firing rate, and thus can be handled by our DSR framework. In contrast, if firing rate changes rapidly in most bins, our inference method would attribute such ongoing rapid changes to spiking irregularity.

We present these results in new Supplementary Note 1.2, new Supplementary Figs. 1–3, on lines 139–144 in revised Results, and on lines 586–594 in revised Methods.

2. *“In all areas, neurons with high firing rates had low spiking irregularity: among 25% neurons with the highest firing rates, virtually...” The refractory period can substantially decrease the variability in spike counts, by negatively correlating spike probabilities. Some consideration and discussion of the effect of absolute and relative refractory periods would be useful to tie this approach to the biophysics of single neurons.*

Reply:

The refractory period can indeed contribute to the negative correlation between the mean firing rate and spiking irregularity. Prompted by the reviewer’s comment, we performed two additional analyses to verify that the refractory period is not the sole source of the negative correlation observed in the data (Fig. 4). First, we estimated the mean firing rate above which the refractory period substantially impacts the spiking irregularity. Then, we systematically removed high firing-rate neurons from experimental data to verify that the negative correlation between the firing rate and spiking irregularity is not solely driven by spiking regularity of high firing-rate neurons.

First, we estimate the firing rate above which the refractory period significantly impacts spiking irregularity. We consider an integrate-and-fire type neuron that fires a spike when the voltage reaches a threshold V_{th} , upon which the voltage is reset to V_r after a fixed refractory period RP . The neuron receives a presynaptic input spike train following a homogeneous renewal process with a constant firing rate λ_{in} . Each presynaptic spike increments the membrane potential by an amount J . In the stationary regime, we can decompose the postsynaptic interspike interval (ISI) into the refractory period and a random component: $ISI = RP + X$. The probability distribution of random variable X generally depends on parameters such as neuron properties, input statistics, and also RP . Let us assume that $(V_{th} - V_r)/J \gg 1$, meaning that the neuron needs to receive $n \gg 1$ presynaptic spikes to generate the next spike after

a reset. We can then write $X = t_0 + \sum_{i=1}^{n-1} t_i$, where t_0 is time between the end of RP and arrival of the first presynaptic spike, and t_i is the ISI between the i -th and $(i+1)$ -th presynaptic spikes. Since the input process is renewal, only t_0 in X is influenced by the refractory period, and X is approximately independent of RP for large n . For constant input firing rate, we can compute the spiking irregularity using ISI moments in real time, instead of

operational time, to obtain $\varphi = \frac{\sigma_{ISI}^2}{\mu_{ISI}^2} = \frac{\sigma_X^2}{(RP + \mu_X)^2} = \frac{CV_X^2}{(\frac{RP}{\mu_X} + 1)^2}$. Here CV_X is the coefficient of variation of X , and the average firing rate of the neuron is $fr = \frac{1}{RP + \mu_X}$. This equation shows that spiking irregularity decreases with

increasing RP . In the limit $\mu_X \gg RP$ (equivalently, $RP \ll \frac{1}{fr}$), the refractory period has negligible effect and $\varphi \approx CV_X^2$. In the opposite limit $\mu_X \ll RP$ (equivalently, $RP \approx \frac{1}{fr}$), the refractory period dominates and $\varphi \approx 0$. In between these two extremes, both RP and X jointly determine the spiking irregularity. The refractory period (including the duration of the action potential) of neurons typically ranges from approximately 4 to 12 ms [1]. Hence, the firing rate above which the refractory period dominates spiking irregularity ranges from $\frac{1}{12\text{ ms}} \approx 83\text{ Hz}$ to $\frac{1}{4\text{ ms}} \approx 250\text{ Hz}$.

Second, we examined the potential effect of the refractory period on the negative correlation between the firing rate and spiking irregularity in experimental data. The correlation remained significant when restricting analysis to neurons with firing rates below 83 Hz (LIP $r = -0.51$, $p = 0.0001$; V4 $r = -0.24$, $p = 0.0001$) and even below 50 Hz, typical of experimental data (LIP $r = -0.48$, $p = 0.0015$; V4 $r = -0.22$, $p = 0.0002$). Therefore, the negative correlation between firing rate and spiking irregularity does not arise solely from the refractory period effects of high firing-rate neurons.

We describe these analyses on lines 260–261 in revised Results and in new Supplementary Note 1.5.

[1] Sardi, Shira, et al. "Long anisotropic absolute refractory periods with rapid rise times to reliable responsiveness." *Physical Review E* 105.1 (2022): 014401.

3. I think it would be more useful to have Figure 3 C as a scatter plot. At least for phi and spiking irregularity.

Reply:

We added the scatter plot as an additional new panel in Fig. 3d.

4. It would be useful to discuss this approach, and the results shown across different brain areas, in the context of these older papers (both of which were referenced):

Maimon, G. & Assad, J. A. Beyond Poisson: increased spike-time regularity across primate parietal cortex. *Neuron* 62, 426–440 (2009).

Shinomoto, S. et al. Relating neuronal firing patterns to functional differentiation of cerebral cortex. *Plos Comput Biol* 5 (2009).

Reply:

We thank the review for this excellent suggestion to further discuss our results and approach within the context of these older papers. Consistent with these previous studies, we find that spiking irregularity decreases along the cortical hierarchy. However, previous studies used metrics to quantify spiking irregularity in data lacking a generative model, which left it unclear what quantity these metrics estimate and whether different methods estimate the same or different quantities. Therefore, it was also not possible to assess the accuracy of these methods, leaving uncertainty about the reliability of derived conclusions. In addition, these irregularity metrics cannot be integrated into models of single-trial neural dynamics, as they lack a connection to any generative model of spiking activity. Our work overcomes all these limitations by introducing a mathematical definition of spiking irregularity as a parameter within a generative model, which enables us to verify the accuracy of our estimation method and opens a possibility of integrating spiking irregularity into models of single-trial neural dynamics beyond the standard Poisson assumption.

We added this discussion on lines 409–419 in revised Discussion.

Also, it would be useful to add some discussion of how these measures of variability relate to the time-scale estimates that other groups have been looking at:

<https://pubmed.ncbi.nlm.nih.gov/33431695/>

<https://pubmed.ncbi.nlm.nih.gov/32839338/>

Reply:

We thank the reviewer for the interesting suggestion to relate spiking irregularity to intrinsic neural timescales. Intrinsic timescales are defined by the exponential decay rate of the autocorrelation function of spiking activity. These timescales typically range from tens to several hundred milliseconds, reflecting primarily slow firing-rate dynamics rather than spiking irregularity. While intrinsic timescales systematically increase along the cortical hierarchy [1–4], our results and previous studies [5,6] show that spiking irregularity decreases from visual to association to motor cortical areas. Thus, firing-rate timescales and spiking irregularity follow inverse gradients that align with the functional specialization of cortical areas. A precise Bayesian estimation method [7] revealed that spiking activity in the primate visual cortex unfolds on at least two timescales: a fast ~5 ms timescale and a slow ~100 ms timescale [8]. The millisecond range of the fast timescale may partly reflect spiking irregularity. Moreover, the slow—but not the fast—timescale increased during selective attention [8], consistent with our observation that spiking irregularity remains invariant while firing-rate fluctuations are stabilized during attention.

We added this discussion on lines 420–432 in revised Discussion.

- [1] Murray et al., *Nature Neuroscience* (2014).
- [2] Song, Min, et al., *Proceedings of the National Academy of Sciences* (2024).
- [3] Spitmaan et al., *Proceedings of the National Academy of Sciences* (2020).
- [4] Rossi-Pool et al., *Proceedings of the National Academy of Sciences* (2021).
- [5] Maimon, G., and J. A. Assad., *Neuron* (2009).
- [6] Shinomoto, S., et al., *PLoS Computational Biology* (2009).
- [7] Zeraati, R., et al., *Nature Computational Science* (2022).
- [8] Zeraati, Roxana, et al., *Nature Communications* (2023).

Reviewer #3:

This manuscript presents a new methodology to partition the total spiking variability of neurons into firing rate fluctuations and spiking irregularity. Previous works have addressed this issue with alternative methods, and the manuscript shows that the new method is superior because it can correctly parse firing rate fluctuations across trials and across time within trial. The manuscript presents the mathematical derivations supporting their method, which includes a smart transformation to an "operational" dimension, and the mathematical comparison with existing methods. The method is validated against surrogate data from diffusion models where the contributions of the various forms of variability is known. It is also validated using intracellularly recorded neurons, in a clever approach that uses voltage traces as a proxy to derive the latent firing rate variability. Then, the method is applied to three datasets of extracellularly recorded neurons in different cortical areas during various laboratory tasks. Several results in the literature are replicated, such as changes in neural variability across the cortical hierarchy, or depending on behavioral state. Finally, a computational network model is used to show that results compatible with the observations in the datasets can be obtained from spiking neural network simulations.

In my opinion, this manuscript presents a relevant new methodology to analyze neural recordings in behaving animals and provide a more rigorous way to separate the sources of neural variability in order to facilitate the interpretation of data. The mathematical derivations are elegant and clearly presented, and their strength is nicely demonstrated with a variety of validation methods: from analytical comparison with competitor methods, to direct validation and comparison against ground truth surrogate data, and to an elegant validation with intracellularly recorded voltage traces. I found more shallow the application of the method to existing datasets and network simulations.

Reply:

We thank the reviewer for recognizing the strength of our computational framework and the significance of its validation with intracellular voltage recordings. We also appreciate the thoughtful comments, which prompted us to make three key improvements to the manuscript. First, we simulated two additional spiking network models that

exhibit substantial firing rate fluctuations and reveal complementary mechanisms underlying spiking irregularity. Second, we showed in V4 data and in a spiking network model that attention stabilizes firing rates over longer timescales without affecting spiking irregularity of single neurons, providing new insights into mechanisms of selective attention. Finally, we clarified the conceptual advance of our mathematical framework over previous methods for quantifying spiking irregularity. Together, these revisions strengthen the applications of our framework to data and network models and enhance the presentation clarity of our results in the paper.

My major concerns are:

1) There is no clear example where the application of this new method proves crucial to gain new insight into brain function. In particular, previous methods already concluded that neural variability declines in higher cortical areas (Maimon et al Neuron 2009). The overall message from this section is that the method is able to replicate previous results, but does not offer a specific advantage to correct previous misinterpretations.

Reply:

While the conceptual idea of partitioning spiking variability into firing rate and spiking irregularity has a long-standing history, with several methods proposed for estimating spiking irregularity in data, our work presents two key advances beyond these previous studies.

First, we introduce a rigorous mathematical framework formalizing the idea of partitioning variability, which allows us to define spiking irregularity precisely as a parameter in our doubly stochastic renewal model. Previous work shows that any partitioning of variability is ambiguous without an underlying generative model (Amarasingham et al., *PNAS*, 2015). While previous studies defined various metrics to quantify spiking irregularity in data, producing sensible results (e.g., Maimon et al., *Neuron* 2009), the absence of a generative model left it unclear what quantity these metrics estimate and whether different methods estimate the same or different quantities. Therefore, it was also not possible to assess the accuracy of these methods, leaving uncertainty about the reliability of derived conclusions. In addition, irregularity metrics cannot be integrated into models of single-trial neural dynamics, as they lack a connection to any generative model of spiking activity. Our work overcomes all these limitations by introducing a mathematical definition of spiking irregularity as a parameter within a generative model, which enables us to verify the accuracy of our estimation method and opens a possibility of integrating spiking irregularity into models of single-trial neural dynamics beyond the standard Poisson assumption.

Second, we confirm that our doubly stochastic renewal model aligns with the biophysical properties of neural circuits. This connection has not been tested despite the wide use of doubly stochastic point processes for modeling spiking activity. In fact, the reliable spiking of cortical neurons in response to time-varying inputs (Pattadkal et al., *bioRxiv* 2024) calls into question whether stochastic spike generation models are even compatible with circuit biophysics. Therefore, we validate our doubly stochastic renewal model using intracellular voltage recordings and simulations of spiking network models, which provides a justification for the widespread use of doubly stochastic models in single-trial spike analysis.

We clarify these points on lines 393–402 and 409–417 in revised Discussion.

On the other hand, the invariance of neural variability with attentional conditions could be interesting but is not fully discussed.

Reply:

We thank the reviewer for encouraging us to further explore the invariance of spiking irregularity with attention. While it is well established that total spiking variability, as measured by the Fano factor (FF), decreases during attention, it has remained unknown whether the FF decrease results from a reduction in spiking irregularity, firing-rate fluctuations, or both. Our approach revealed that spiking irregularity remained invariant across attention conditions (Fig. 5b) and the decrease in total variability resulted entirely from a reduction in trial-to-trial firing-rate fluctuations. Thus, attention stabilizes firing rates over longer timescales without affecting spiking irregularity of single neurons. This result suggests that attention enhances information transmitted through the firing rates rather than precise spike patterns, providing new insights into mechanisms of selective attention. In addition, the invariance of spiking irregularity provides tight constraints for biophysical network models of attention.

Following the reviewer's suggestion, we further tested whether a previously proposed biophysical circuit model of attention (Huang et al., *Neuron* 2019) satisfies the constraint of invariant spiking irregularity revealed by our analyses (Fig. 5b). We simulated the three-layer spiking network model, in which attentional modulation is implemented as a static depolarizing current injected into inhibitory neurons in the third layer (Huang et al., *Neuron* 2019). It has been shown previously that this model replicates the reduction in FF during attention, and we tested whether the FF reduction in the model resulted from changes in spiking irregularity or firing-rate fluctuations. The spiking irregularity in the model was invariant across attention conditions (Fig. 6e), consistent with our observations in experimental data, therefore supporting the proposed circuit mechanism of attentional modulation. Our method is uniquely suited to detect this dissociation, providing new insight into how attention modulates neural variability.

We present these results in the new Fig. 6e and on lines 281–291 and 359–371 in revised Results.

The increase of neural variability in the decision period relative to the fixation period observed is presented in the Discussion as contrary to existing literature (citing perceptual, motor and attention tasks), but not in relation to more similar decision and memory tasks, which go in the same direction (Churchland et al. Neuron 2011; Compte et al. J Neurophysiol 2003).

Reply:

We thank the reviewer for indicating that this text in the original manuscript was ambiguous, leading to some confusion. Churchland et al. (*Neuron* 2011) observed an increase in *firing-rate* variability (on long timescales) through the decision period. Their analysis assumed that spiking irregularity (on short timescales) is a neuron-specific constant, invariant across time and stimulus conditions, without testing this assumption in data. In contrast, we found that the spiking irregularity was invariant in many cases, such as across different attention states or decision difficulties, but changed between different epochs of the decision-making task in PMd.

We clarified this point on lines 443–454 in the revised Discussion.

2) It is unclear how the model simulations help to support the new methodology, especially since the simulations do not incorporate variability in the firing rate (see e.g. Ostojic Nat Neurosci 2014, 10.1038/nn.3658) so there is not a problem of variability attribution.

Reply:

We thank the reviewer for this comment, which prompted us to test whether the two-layer balanced network model (Huang et al., *Neuron* 2019) used in our original manuscript exhibits firing-rate fluctuations. Indeed, while balanced networks are well known to generate variable spiking activity, it is unclear what fraction of this variability can be attributed to spiking irregularity versus firing-rate fluctuations. As the reviewer correctly predicted, in the two-layer model, CV^2_{raw} of ISIs in real time was very close to ϕ , indicating that variability arises almost entirely from spiking irregularity without pronounced firing-rate fluctuations. We therefore followed the reviewer's suggestion and replaced this model with two other models in the revised manuscript: 1) the three-layer balanced network model with spatial connectivity (Huang et al., *Neuron* 2019), and 2) the random balanced network model with strong synaptic couplings (Ostojic, *Nat Neurosci* 2014). We verified that these both models generate firing-rate fluctuations, such that CV^2_{raw} of ISIs in real time is substantially larger than ϕ (three-layers network: CV^2_{raw} is 17% larger than ϕ , paired t-test, $p < 10^{-10}$, $n=50,000$; random balanced network model with $J=0.4$, CV^2_{raw} is 10% larger than ϕ , paired t-test, $p < 10^{-10}$, $n=8,000$).

These two models offer complementary advantages for our study. The three-layers network (Huang et al., *Neuron* 2019) allows us to test the invariance of spiking irregularity with attention (see our response to question 1 above). In addition, this model has a varying number of excitatory and inhibitory connections across neurons, which allows us to show that spiking irregularity depends on the balance between excitatory and inhibitory inputs received by each neuron (Fig. 6d). In contrast, in the random balanced network model (Ostojic, *Nat Neurosci* 2014), each neuron receives exactly the same number of excitatory and inhibitory connections, allowing us to demonstrate that recurrent dynamics provide an alternative mechanism for generating diverse spiking irregularity across neurons, beyond

differences in their synaptic input balance. In addition, this model generates variability solely from its internal dynamics, whereas the three-layer model includes a Poisson layer as an external source of variability.

We present these results in revised Fig. 6, new Supplementary Note 1.6, new Supplementary Fig. 5 and on lines 342–346 and 359–371 in revised Results.

Also, the fact that balanced excitatory-inhibitory networks with probabilistic connectivity structures have a broad distribution of firing rates and neural variability, comparable to in situ data, is known (van Vreeswijk and Sompolinsky, Science 1996; Neural Comput 1998; Roxin et al. J Neurosci 2011).

Reply:

While it is well known that balanced excitatory-inhibitory networks generate diverse and variable spiking activity, it has not been tested whether this variability arises from spiking irregularity or firing-rate fluctuations. Using our approach, we 1) provide new insights into dynamics of balanced networks through analyses of spiking irregularity in the model from Ostojic (*Nat Neurosci* 2014) across a range of synaptic strength parameters J , and 2) suggest what factors can enable better match of these models to experimental data.

We find that in the classical weak-coupling asynchronous regime, which occurs for small J , neurons exhibit sub-Poisson spiking irregularity with narrow distribution of ϕ across neurons (Supplementary Fig. 5a,c). For larger J , the distribution broadens and the average ϕ increases, such that neurons exhibit super-Poisson irregularity with diverse ϕ in the strong-coupling regime (Supplementary Fig. 5b,c). Since each neuron in this model receives the same number of excitatory and inhibitory connections, the diverse spiking irregularity arises solely from recurrent dynamics in the heterogeneous asynchronous regime, providing a mechanism distinct from the heterogeneity in synaptic input balance (Fig. 6b-d). However, this model generates sub-Poisson spiking irregularity only with narrow ϕ distribution, whereas the LIP and PMd data exhibit a broad range of sub-Poisson spiking irregularity across neurons (Fig. 4a). These results suggest that heterogeneity in incoming synaptic connections in random balanced networks may be necessary to produce the diverse sub-Poisson spiking irregularity, similar to our LIP and PMd data.

While previous studies reported that total spiking variability increases with synaptic strength J [1,2], our framework enables us to determine whether this increase results from increased spiking irregularity or firing-rate variability. We found that both spiking irregularity and firing rate variability increase with synaptic strength, indicating that stronger synaptic connections elevate spiking variability on both short and long timescales (Supplementary Fig. 5c,d).

We present these results in new Supplementary Note 1.6, new Supplementary Fig. 5, and on lines 342–346 in revised Results.

[1] Ostojic, "Two types of asynchronous activity in networks of excitatory and inhibitory spiking neurons." *Nature Neuroscience* 17.4 (2014): 594-600.

[2] Brunel, "Dynamics of sparsely connected networks of excitatory and inhibitory spiking neurons." *Journal of computational neuroscience* 8 (2000): 183-208.

On the other hand, the manipulations to alter neural irregularity in the network are over-simplified: considering that a focus is on changes in conductance it would appear necessary to consider conductance-based synapses to understand the modulations expected in more realistically connected networks (lines 393-394).

Reply:

We agree that this analysis abstracts many biological details. Therefore, we moved it to new Supplementary Note 1.7 and new Supplementary Fig. 6, referenced on lines 353–357 in revised Results.

3) Instead, the modeling frustrates the expectations of the reader, since at several points in the manuscript it is speculated that changes in spiking irregularity in the cortical hierarchy could relate to functional specialization, but this is never tested in simulations. Instead of spontaneous activity without any specific task structure in the simulation, known tasks could be simulated to address how changes in network dynamics related to specific tasks could modulate

irregularity. The model could generate function by putting it into and out of some attractor dynamics (e.g. Hansel and Mato, *J Neurosci* 2013, 10.1523/JNEUROSCI.3455-12.2013) or applying some attentional modulation (e.g. Huang et al. *Neuron* 2019 10.1016/j.neuron.2018.11.034).

Reply:

We absolutely agree that analyzing spiking irregularity in a model simulating a cognitive function will significantly strengthen our results. Therefore, following the reviewer’s suggestion, we simulated a previously proposed biophysical circuit model of attention (Huang et al., *Neuron* 2019). This model is a three-layer spiking neural network, in which attentional modulation is implemented as a static depolarizing current injected into inhibitory neurons in the third layer (Huang et al., *Neuron* 2019). It has been shown previously that this model replicates the reduction in FF during attention. We tested whether the FF reduction in the model resulted from changes in spiking irregularity or firing-rate fluctuations. We found that the spiking irregularity in the model was invariant across attention conditions (Fig. 6e), which is consistent with our analysis of experimental data during a spatial attention task (Fig. 5b). These results provide further support for the proposed circuit mechanism of attentional modulation in the spiking network model.

We present these results in the new Fig. 6e and on lines 359–371 in revised Results.

Minor comments:

1) line 209 citation to ref. 80 seems to forget the classic studies that first proposed this calculation: Ricciardi 1977, Amit and Tsodyks 1981, Amit and Brunel 1997

Reply:

We thank the reviewer for this reminder. We revised this paragraph based on point 3 of referee 1. In the revised version, we do not cite the reference number 80 anymore.

2) line 692: please define x_i , it is necessary to understand the following equation. Is it $x_i = i/N_\alpha$?

Reply:

All neurons (N_f , N_e , and N_i) are uniformly distributed on a unit square where the position (x_i, y_i) of neuron $1 \leq i \leq N_\alpha$ is $0 \leq x_i = \frac{1}{\sqrt{N_\alpha-1}} * \text{mod}(i - 1, \sqrt{N_\alpha}), y_i = \frac{1}{\sqrt{N_\alpha-1}} * \text{floor division}(i - 1, \sqrt{N_\alpha}) \leq 1$.

We added this text on lines 748–750 in revised Methods.

3) line 693: this probability function is only approximately properly normalized if $\sigma \ll 1$. Also, could the summation be restricted to be from $k=-1$ to $k=1$, when $\sigma \ll 1$ and $-1 < r < 1$?

Reply:

Thanks to the reviewer’s feedback, we noticed and corrected a typo: “k” needs to change to “2k” in this function. Here we show that this function is correctly normalized for any value of sigma. We believe the confusion arose because the domain of the function is $[-1,1]$ and not $[-\infty, +\infty]$, which is the cases since $-1 \leq x_i - x_j \leq 1$. To show that the function is correctly normalized, we integrate it over its domain:

$$\int_{-1}^1 f(r, \sigma) dr = \int_{-1}^1 \frac{1}{\sqrt{2\pi\sigma}} \sum_{k=-\infty}^{\infty} \exp\left[-\frac{(r+2k)^2}{2\sigma^2}\right] dr = \sum_{k=-\infty}^{\infty} \left(\int_{-1}^1 \frac{1}{\sqrt{2\pi\sigma}} \exp\left[-\frac{(r+2k)^2}{2\sigma^2}\right] dr \right)$$

$$= \sum_{k=-\infty}^{\infty} \left(\int_{-1+2k}^{1+2k} \frac{1}{\sqrt{2\pi\sigma}} \exp\left[-\frac{(r)^2}{2\sigma^2}\right] dr \right) = \int_{-\infty}^{\infty} \frac{1}{\sqrt{2\pi\sigma}} \exp\left[-\frac{(r)^2}{2\sigma^2}\right] dr = 1.$$

We revised the typo on line 753 in revised Methods.

4) line 702: $J_{kj}^\alpha \beta$ has never been defined. Same for μ_β (which is confusing because μ has been used for something else before), g_β , Δ_T , V_T , E_L

Reply:

$J_{kj}^{\alpha\beta}$ is synaptic weight from neuron k to neuron j , where α indicates the type of neuron k and β indicates the type of neuron j . μ_{β} is the static current injected into neurons of type β . g_{β} is the membrane conductance of neurons of type β . C_m is the membrane capacitance. E_L is the resting potential. V_T is the spike initiation threshold, the voltage level at which the neuron begins to exhibit rapid depolarization, leading to spike generation. When the membrane potential approaches or exceeds V_T , the exponential term in the EIF model, becomes substantial, causing a sharp increase in the membrane potential and triggering a spike. Δ_T is the sharpness parameter, which controls the steepness of the exponential rise in the membrane potential as the neuron approaches the threshold V_T . A larger Δ_T results in a more gradual spike initiation, while a smaller Δ_T leads to a sharper, more abrupt spike.

We added these definitions on lines 761–766 and 769–771 in the revised Methods.

(Remarks on code availability):

The source code to reproduce the results of this study will be made available on GitHub upon publication.

Reply:

Yes, we are preparing the code for public release on GitHub at the time of publication, including refactoring and creating clear documentation to ensure usability and reproducibility.

Reviewer #4:

Spike rates have long been considered the fundamental currency of neural computations. The noisiness of neuronal spiking is traditionally modeled as a Poisson process. But the observed spiking patterns in many areas of the brain, especially in association and premotor areas, significantly deviate from the predictions of a Poisson process. And yet, there is no established model to accurately partition variance arising from underlying rates vs. the variance that comes from the spike generation process itself. The current manuscript provides an impressive mathematical framework for accomplishing that.

This study is a tour de force:

- *The mathematical approach is elegant and well thought out,*
- *The model is thoroughly validated at multiple levels — from synthetic data; from intracellular recordings; from single unit data from sensory, association and motor cortices; and from in silico modeling*
- *The authors present novel insights from this modeling approach — establishing how spike generation process varies between sensory, association and motor cortices; and how it can even vary for the same neuron across task epochs*
- *The narrative is very well written and easy to follow*
- *A clear recipe is provided for the field to implement this model with their own data.*
- *The approach presented has the potential to be widely used in systems neuroscience*

Reply:

We sincerely thank the reviewer for the enthusiastic evaluation and for recognizing the significance of our work.

I have no major concerns about the manuscript. Below, I list a few minor suggestions that the authors could consider for improving the manuscript:

Figure 3c: The description is hard to follow. It would be helpful if this panel is introduced by a topic sentence. Also, a horizontal line at 1 could help readers make a quick visual assessment of how close ϕ is to 1 for each neuron.

Reply:

We added this topic sentence to the caption of Fig. 3c “To independently estimate the spiking irregularity of a neuron, we map its spikes to the operational time using the instantaneous firing rate computed from the subthreshold membrane potential at each time using the fitted voltage-to-firing-rate relationship in b.” We also added the horizontal line at 1 to make the visual assessment easier.

Ln 227: The claim that the spiking becomes “... progressively more ...” regular is debatable. There appears to be a bimodality to the LIP ϕ distribution, which places subpopulations of LIP neurons on either side of PMd ϕ distribution.

Reply:

We agree with this comment. We changed this text to read “The average spiking irregularity of neurons was slightly super-Poisson in V4 ($\phi = 1.22 \pm 0.03$, mean \pm std across neurons) and became sub-Poisson and more regular in LIP ($\phi = 0.68 \pm 0.04$) and PMd ($\phi = 0.51 \pm 0.02$).”

We revised this text on lines 238–239 in revised Results.

Ln 242: Similarly, the claim that virtually no neurons reside in the top right corner of Figure 4b can be softened.

Reply:

We changed this text to read “among 25% neurons with the highest firing rates, only few neurons had ϕ within the top 25%”. We also changed the caption of Fig. 4b to read “Only few neurons had ϕ within the top 25%”.

We revised the text on lines 253–255 in revised Results.

Ln 63/64: It is hard to distinguish $\mathbf{\lambda}$ from λ , especially in print.

Reply:

We agree that clear notation is important. In this case, we prefer to retain the current notation, as the distinction between a random variable $\mathbf{\lambda}$ and its sample λ follows a common convention, which we apply consistently to all random variables in the manuscript. If accepted for publication, the manuscript will be professionally typeset, and we expect that the final typesetting will make the bold and non-bold symbols more easily distinguishable.

Point-by-point responses.

Reviewer #1:

The authors have addressed all my concerns and I now fully support the publication of this important paper in Nature Communications.

Reviewer #2:

The authors have addressed my concerns. I have no further comments.

Reviewer #3:

I am fully satisfied with the revisions of this manuscript. I congratulate the authors for this excellent paper.

Reviewer #4:

The authors have comprehensively addressed all reviewer comments. My assessment of the rigor and potential impact of the original manuscript itself was already highly positive and the revisions further strengthen both. I have no remaining concerns.

Reply:

We thank all reviewers again for their thoughtful comments, which helped to strengthen our results and significantly improve the quality of the manuscript. We are pleased that our revisions have satisfactorily addressed all concerns, and we thank the reviewers once more for recognizing the broad significance of our work.